# Gene regulatory network reconstruction using single-cell RNA sequencing of barcoded genotypes in diverse environments

Christopher A Jackson[1,2†], Dayanne M Castro[2†], Giuseppe-Antonio Saldi[2], Richard Bonneau[1,2,3,4,5‡*], David Gresham[1,2‡*]

[1]Center For Genomics and Systems Biology, New York University, New York, United States; [2]Department of Biology, New York University, New York, United States; [3]Courant Institute of Mathematical Sciences, Computer Science Department, New York University, New York, United States; [4]Center For Data Science, New York University, New York, United States; [5]Flatiron Institute, Center for Computational Biology, Simons Foundation, New York, United States

**Abstract** Understanding how gene expression programs are controlled requires identifying regulatory relationships between transcription factors and target genes. Gene regulatory networks are typically constructed from gene expression data acquired following genetic perturbation or environmental stimulus. Single-cell RNA sequencing (scRNAseq) captures the gene expression state of thousands of individual cells in a single experiment, offering advantages in combinatorial experimental design, large numbers of independent measurements, and accessing the interaction between the cell cycle and environmental responses that is hidden by population-level analysis of gene expression. To leverage these advantages, we developed a method for scRNAseq in budding yeast (*Saccharomyces cerevisiae*). We pooled diverse transcriptionally barcoded gene deletion mutants in 11 different environmental conditions and determined their expression state by sequencing 38,285 individual cells. We benchmarked a framework for learning gene regulatory networks from scRNAseq data that incorporates multitask learning and constructed a global gene regulatory network comprising 12,228 interactions.

\*For correspondence:
bonneau@nyu.edu (RB);
dgresham@nyu.edu (DG)

[†]These authors contributed equally to this work
[‡]These authors also contributed equally to this work

**Competing interests:** The authors declare that no competing interests exist.

## Introduction

Elucidating relationships between genes, and the products they encode, remains one of the central challenges in experimental and computational biology. A gene regulatory network (GRN) is a directed graph in which regulators of gene expression are connected to target gene nodes by interaction edges. Regulators of gene expression include transcription factors (TF) which can act as activators and repressors, RNA binding proteins, and regulatory RNAs. Identifying regulatory relationships between transcriptional regulators and their targets is essential for understanding biological phenomena ranging from cell growth and division to cell differentiation and development (*Davidson, 2012*). Reconstruction of GRNs is required to understand how gene expression dysregulation contributes to cancer and complex heritable diseases (*Barabási et al., 2011*; *Hu et al., 2016*).

Genome-scale methods provide an efficient means of identifying gene regulatory relationships. Efforts of the past two decades have resulted in the development of a variety of experimental and computational methods that leverage advances in technology and machine learning for constructing GRNs. Previously, we developed a method for inferring transcriptional regulatory networks based on regression with regularization that we have called the Inferelator (*Bonneau et al., 2006*;

**eLife digest** Organisms switch their genes on and off to adapt to changing environments. This takes place thanks to complex networks of regulators that control which genes are actively 'read' by the cell to create the RNA molecules that are needed at the time. Piecing together these networks is key to fully understand the inner workings of living organisms, and how to potentially modify or artificially create them.

Single-cell RNA sequencing is a powerful new tool that can measure which genes are turned on (or 'expressed') in an individual cell. Datasets with millions of gene expression profiles for individual cells now exist for organisms such as mice or humans. Yet, it is difficult to use these data to reconstruct networks of regulators; this is partly because scientists are not sure if the computational methods normally used to build these networks also work for single-cell RNA sequencing data.

One way to check if this is the case is to use the methods on single-cell datasets from organisms where the networks of regulators are already known, and check whether the computational tools help to reach the same conclusion. Unfortunately, the regulatory networks in the organisms for which scientists have a lot of single-cell RNA sequencing data are still poorly known. There are living beings in which the networks are well characterised – such as yeast – but it has been difficult to do single-cell sequencing in them at the scale seen in other organisms.

Jackson, Castro et al. first adapted a system for single-cell sequencing so that it would work in yeast. This generated a gene expression dataset of over 40,000 yeast cells. They then used a computational method (called the Inferelator) on these data to construct networks of regulators, and the results showed that the method performed well. This allowed Jackson, Castro et al. to start mapping how different networks connect, for example those that control the response to the environment and cell division. This is one of the benefits of single-cell RNA methods: cell division for example is not a process that can be examined at the level of a population, since the cells may all be at different life stages. In the future, the dataset will also be useful to scientists to benchmark a variety of single cell computational tools.

*Ciofani et al., 2012*). This method takes as inputs gene expression data and sources of prior information, and outputs regulatory relationships between transcription factors and their target genes that explain the observed gene expression levels. Subsequent work has enhanced this approach by selecting regulators for each gene more effectively (*Madar et al., 2010*), incorporating orthogonal data types that can be used to generate constraints on network structure (*Greenfield et al., 2013*), and explicitly estimating latent biophysical parameters including transcription factor activity (*Arrieta-Ortiz et al., 2015*; *Fu et al., 2011*) and mRNA decay rates (*Tchourine et al., 2018*). We have successfully applied this approach to construct GRNs from gene expression data acquired from variation across time, conditions, and genotypes in microbes (*Arrieta-Ortiz et al., 2015*; *Tchourine et al., 2018*), plants (*Wilkins et al., 2016*), and mammalian cells (*Ciofani et al., 2012*; *Miraldi et al., 2019*).

Recently, single-cell RNA sequencing (scRNAseq) has exploded in popularity with the development of droplet systems for rapid encapsulation and labeling of thousands of cells in parallel. The DROP-seq system (*Macosko et al., 2015*) based on bead capture, and the inDrop (*Zilionis et al., 2017*) and 10x Genomics (*Zheng et al., 2017*) systems based on hydrogel beads, provide a facile means of generating RNA sequencing data for tens of thousands of individual cells. Although scRNAseq has primarily been used for defining cell types and states, this technology holds great potential for efficient construction of GRNs (*Hwang et al., 2018*). By combining genetic perturbation of transcriptional regulators using CRISPR/Cas9 with scRNAseq, mixtures of genetic perturbations can be assayed in a single reaction (*Adamson et al., 2016*; *Dixit et al., 2016*; *Jaitin et al., 2016*). This approach, known as Perturb-seq, presents a new opportunity for efficiently inferring GRNs from thousands of individual cells in which different regulators have been disrupted. There are considerable advantages in both scalability and detection of intra-sample heterogeneity with Perturb-seq, but quantifying the the effectiveness of CRISPR/Cas9 targeting in individual cells and distinguishing gene expression variability from noise inherent to mRNA undersampling in scRNAseq (*Brennecke et al., 2013*; *Grün et al., 2014*) present technical challenges. Computational methods

to take advantage of scRNAseq data for inferring GRNs are under active development (*Aibar et al., 2017*; *Chan et al., 2017*; *van Dijk et al., 2018*). However, benchmarking these methods is difficult; in the absence of a known GRN, model performance is often estimated using simulated data (*Chen and Mar, 2018*), and issues regarding the appropriate experimental and computational approaches to GRN construction from scRNAseq data remain unresolved.

The budding yeast *Saccharomyces cerevisiae* is ideally suited to constructing GRNs from experimental data and benchmarking computational methods. Decades of work have provided a plethora of transcriptional regulatory data comprising functional and biochemical information (*de Boer and Hughes, 2012*; *Teixeira et al., 2018*). As a result, yeast is well suited to constructing GRNs using methods that leverage the rich available information and for assessing the performance of those methods by comparison to experimentally validated interactions (*Ma et al., 2014*; *Tchourine et al., 2018*). Budding yeast presents several technical challenges for single cell analysis, and as a result scRNAseq methods for budding yeast reported to date (*Gasch et al., 2017*; *Nadal-Ribelles et al., 2019*) yield far fewer individual cells (~$10^2$) than are now routinely generated for mammalian studies (>$10^4$). The limitations of existing scRNAseq methods for budding yeast cells limits our ability to investigate eukaryotic cell biology as many signaling and regulatory pathways are highly conserved in yeast (*Carmona-Gutierrez et al., 2010*; *Gray et al., 2004*), including the Ras/protein kinase A (PKA), AMP Kinase (AMPK) and target of rapamycin (TOR) pathways (*González and Hall, 2017*; *Loewith and Hall, 2011*). However, recent work has successfully established single cell sequencing in the fission yeast *Schizosaccharomyces pombe* (*Saint et al., 2019*).

In budding yeast, the TOR complex 1 (TORC1 or mTORC1 in human) coordinates the transcriptional response to changes in nitrogen sources (*Godard et al., 2007*; *Rødkaer and Faergeman, 2014*). Controlling this response are four major TF groups, which are regulated by diverse post-transcriptional processes. The Nitrogen Catabolite Repression (NCR) pathway, which is regulated principally by TORC1, consists of the TFs *GAT1*, *GLN3*, *DAL80*, and *GZF3* (*Hofman-Bang, 1999*), and is responsible for suppressing the utilization of non-preferred nitrogen sources when preferred nitrogen sources are available. Gat1 and Gln3 are localized to the cytoplasm until activation results in relocalization to the nucleus (*Cox et al., 2000*), where they then compete with Dal80 and Gzf3 for DNA binding motifs (*Georis et al., 2009*). The General Amino Acid Control (GAAC) pathway consists of the TF *GCN4* (*Hinnebusch, 2005*), and is responsible for activating the response to amino acid starvation, as detected by increases in uncharged tRNA levels. Gcn4 activity is translationally controlled by ribosomal pausing at upstream open reading frames in the 5' untranslated region (*Mueller and Hinnebusch, 1986*). The retrograde pathway, consisting of the TF heterodimer *RTG1* and *RTG3*, is responsible for altering expression of metabolic and biosynthetic genes in response to mitochondrial stress (*Jia et al., 1997*; *Liao and Butow, 1993*) or environmental stress (*Ruiz-Roig et al., 2012*). The Rtg1/Rtg3 complex is localized to the cytoplasm until activation, upon which they translocate into the nucleus (*Komeili et al., 2000*). The Ssy1-Ptr3-Ssy5-sensing (SPS) pathway (*Ljungdahl, 2009*), consists of the TFs *STP1* and *STP2*, and is responsible for altering transporter expression (*Didion et al., 1998*; *Iraqui et al., 1999*) in response to changes in extracellular environment. Stp1 and Stp2 are anchored to the plasma membrane until the SPS sensor triggers proteolytic cleavage of their anchoring domain and releases them for nuclear import (*Andréasson and Ljungdahl, 2002*).

Construction of GRNs based on the transcription factors in these pathways has had mixed success; the high redundancy of the NCR pathway has proven challenging to deconvolute (*Milias-Argeitis et al., 2016*). The GAAC pathway is more straightforward, although separating direct and indirect regulation remains difficult, even with high-quality experimental data (*Mittal et al., 2017*). As a result, a comprehensive GRN for nitrogen metabolism has remained elusive, despite successes in identifying genes that respond to changes in environmental nitrogen (*Airoldi et al., 2016*) and identification of post-transcriptional control mechanisms that underlie these changes (*Miller et al., 2018*).

Many signalling regulators involved in environmental response interact with cell cycle programs (*Johnston et al., 1977*; *Talarek et al., 2017*), including the TOR pathway (*Zinzalla et al., 2007*); however, how regulation of the mitotic cell cycle and environmentally responsive gene expression is coordinated is unknown. The regulation of nitrogen responsive gene expression in yeast is well-suited to the development of generalizable methods as the degree of TF redundancy, post-transcriptional regulation of TF activity, which precludes straightforward relationships between TF

abundance and target expression, and multifactorial impact on gene expression, including intrinsic and extrinsic processes and stimuli, provide a tractable model system for addressing these challenges in higher eukaryotes.

Here, we have developed a method for scRNAseq in budding yeast using Chromium droplet-based single cell encapsulation (10x Genomics). We engineered TF gene deletions by precisely excising the entire TF open reading frame and introducing a unique transcriptional barcode that enables multiplexed analysis of genotypes using scRNAseq. We pooled 72 different strains, corresponding to 12 different genotypes, and determined their gene expression profiles in 11 conditions using scRNAseq analysis of 38,000 cells. We show that our method enables identification of cells from complex mixtures of genotypes in asynchronous cultures that correspond to specific mutants, and to specific stages of the cell cycle. Identification of mutants can be used to identify differentially expressed genes between genotypes providing an efficient means of multiplexed gene expression analysis. We used scRNAseq data in yeast to benchmark computational aspects of GRN reconstruction, and show that multi-task learning integrates information across environmental conditions without requiring complex normalization, resulting in improved GRN reconstruction. We find that imputation of missing data does not improve GRN reconstruction and can lead to prediction of spurious interactions. Using scRNAseq data, we constructed a global GRN for budding yeast comprising 12,228 regulatory interactions. We discover novel regulatory relationships, including previously unknown connections between regulators of cell cycle gene expression and nitrogen responsive gene expression. Our study provides a generalizable framework for GRN reconstruction from scRNAseq, a rich data set that will enable benchmarking of future computational methods, and establishes the use of droplet-based scRNAseq analysis of multiplexed genotypes in yeast.

## Results

### Engineering a library of Prototrophic, Transcriptionally-Barcoded Gene Deletion Strains

The yeast gene knockout collection (*Giaever et al., 2002*) facilitates pooled analysis of mutants using unique DNA barcode sequences that identify each gene deletion strain, but these barcodes are only present at the DNA level, precluding their use with scRNAseq. Therefore, we constructed an array of prototrophic, diploid yeast strains with homozygous deletions of TFs that control distinct regulatory modules: 1) NCR, 2) GAAC, 3) SPS-sensing, and 4) the retrograde pathway that coordinately control nitrogen-related gene expression in yeast. We engineered eleven different TF knockout genotypes, using six independently constructed biological replicates for each genotype. In addition, we constructed six biological replicates of the wild-type control in which we deleted the neutral HO locus. Genes were deleted using a modified kanMX cassette such that each of the 72 strains contains a unique transcriptional barcode in the 3' untranslated region (UTR) of the G418 resistance gene, that can be recovered by RNA sequencing (*Figure 1*, *Figure 1—figure supplement 1A*). Homozygous diploids were constructed by mating to a strain containing the same TF knockout marked with a nourseothricin drug resistance cassette. On rich media plates, the 72 strains have an approximately wild-type growth; under nutritional stress, some TF knockouts exhibit growth advantages or disadvantages (*Figure 1—figure supplement 1B*).

### Single-Cell RNA sequencing of pooled libraries in diverse growth conditions

ScRNAseq in yeast presents several challenges: cells are small (40–90 $\mu m^3$), enclosed in a polysaccharide-rich cell wall, and contain fewer mRNAs per cell (40 k-60k) than higher eukaryotes. We developed and validated a protocol using the droplet-based 10x genomics chromium platform, and it used it to perform scRNAseq of the pool of TF knockouts in eleven growth conditions that provide a range of metabolic challenges (*Table 1*). In addition to variable nitrogen sources in minimal media with excess [MM] and limiting [NLIM-NH$_4$, NLIM-GLN, NLIM-PRO, NLIM-UREA] nitrogen, some conditions result in fermentative metabolism of glucose in rich media [YPD], and inhibition of the TOR signaling pathway in rich media by the small molecule rapamycin [RAPA]. We also studied conditions that require respiratory metabolism of ethanol in rich [YPEtOH] and minimal media [MMEtOH], and in rich media after sugars had been fully metabolized to ethanol and cells have undergone the

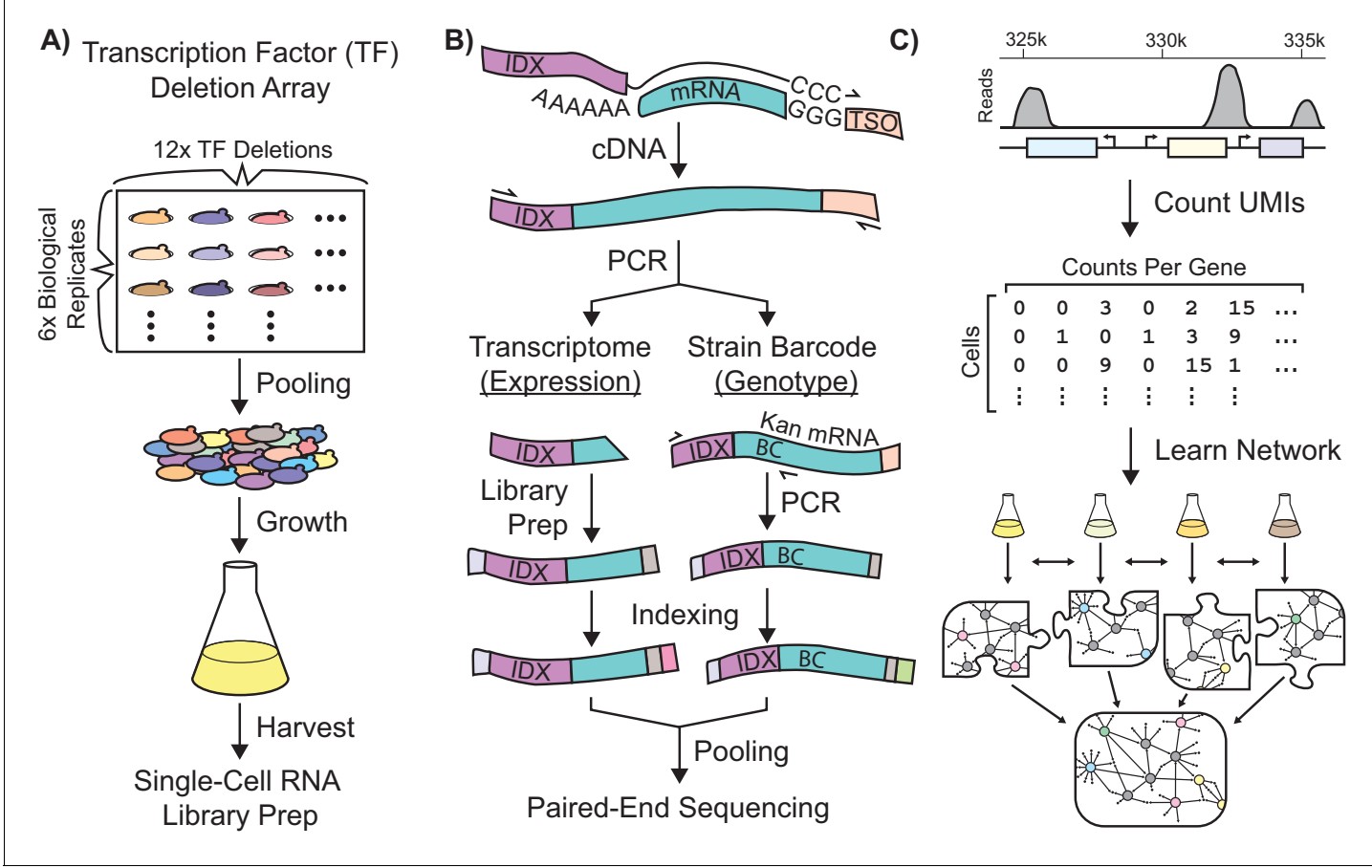

**Figure 1.** Single-Cell RNA-Seq Experimental Workflow in *Saccharomyces Cerevisiae.* Schematic workflows for: (A) Growth of a transcriptionally-barcoded pool of 11 nitrogen metabolism transcription factor (TF) knockout strains and a wild-type control strain each analyzed with six biological replicates (B) Synthesis in microfluidic droplets of single-cell cDNA with a cell-specific index sequence (IDX) attached to the oligo-dT primer, and a common template switch oligo (TSO). cDNA is processed for whole-transcriptome libraries, to quantify gene expression. In parallel, PCR products are amplified containing the genotype-specific transcriptional barcode (BC) encoded on the Kan[R] antibiotic resistance marker mRNA, to identify cell genotype. Expression DNA libraries and PCR products are separately indexed for multiplexed sequencing (C) Processing of single-cell sequencing data using Unique Molecular Identifiers (UMI) into a count matrix which is used to learn a gene regulatory network using multi-task network inference from several different growth conditions.

The online version of this article includes the following figure supplement(s) for figure 1:

**Figure supplement 1.** Strain Construction Workflow and Validation.

diauxic shift [DIAUXY]. We tested two different starvation conditions, carbon [CSTARVE] and nitrogen starvation [NSTARVE]; however, the latter condition did not pass quality control during single-cell transcriptome library preparation and was discarded.

Cells from the eleven different conditions were sequenced and processed using cellranger (10x genomics) and our custom analysis pipeline (fastqToMat0), yielding a digital expression matrix (*Source code 1*) in which each cell is annotated with the environmental growth condition and genotype. Genotype-specific barcodes facilitate identification and removal of droplets that have multiple cells (doublets) by determining cell IDs that have more than one annotated genotype. Using our pool of 72 strains, we detect and remove 98.5% of doublets. PCR artifacts and duplicates are removed using Unique Molecular Identifiers (UMIs) (*Kivioja et al., 2012*) to quantify gene expression as unique transcript reads (counts). Following sequence processing, quality control, removal of doublets, and assigning metadata, we recovered 83,703,440 transcript counts from a total of 38,225 individual cells.

To initially assess the quality of our data, we examined the expression of genes that are characteristic of different metabolic states. Consistent with our expectations, the core fermentative

**Table 1.** Environmental Growth Conditions.

Environmental growth conditions are listed with their respective nitrogen and carbon sources. Yeast Extract + Peptone (YP) is a rich, complex nitrogen source. YP + Dextrose [YPD] is standard yeast rich media. Minimal media contains a standard base of trace metals, vitamins, and salts. All cultures were harvested 4 hr after inoculation, except for the culture harvested after the diauxic shift [DIAUXY], which was harvested 10 hr after inoculation. Rapamycin was added to YPD in the [RAPA] culture 30 min prior to harvest. Specific media formulations are listed in **Supplementary file 1**-Supplemental Table 4.

| Growth condition | Abbrv. | Nitrogen source | Carbon source |
|---|---|---|---|
| Yeast Extract, Peptone, Glucose | YPD | YP | D-Glucose |
| YPD (Harvested after Post-Diauxic Shift) | DIAUXY | YP | D-Glucose |
| YPD + 200 ng/mL Rapamycin | RAPA | YP | D-Glucose |
| Yeast Extract, Peptone, Ethanol | YPEtOH | YP | Ethanol |
| Minimal Media (Glucose) | MMD | 20 mM $(NH_4)_2SO_4$ | D-Glucose |
| Minimal Media (Ethanol) | MMEtOH | 20 mM $(NH_4)_2SO_4$ | Ethanol |
| Nitrogen Limited Minimal Media (with Glutamine) | NLIM-GLN | 0.8 mM L-Glutamine | D-Glucose |
| Nitrogen Limited Minimal Media (with Proline) | NLIM-PRO | 0.8 mM L-Proline | D-Glucose |
| Nitrogen Limited Minimal Media (with NH4) | NLIM-NH4 | 0.8 mM $(NH_4)_2SO_4$ | D-Glucose |
| Nitrogen Limited Minimal Media (with Urea) | NLIM-UREA | 0.8 mM Urea | D-Glucose |
| Carbon Starvation | CSTARVE | 1 mM $(NH_4)_2SO_4$ | None |

(anaerobic) genes *PDC1* and *ENO2* are expressed in cells in fermenting culture conditions only, and the core respirative (aerobic) gene *ADH2* is expressed in cells in respiring culture conditions (*Figure 2A*). The number of cells recovered varies by over an order of magnitude between conditions; stressful conditions of low nitrogen have lower cell yields overall. The yeast stress response is linked to increased resistance to zymolyase digestion (*Nagarajan et al., 2014*), which may be reflected in decreased cell yield during single-cell sequencing. Each of the 72 strains is found in each of the 11 conditions, although the number of each strain and genotype varies by environmental condition (*Figure 2—figure supplement 1A*), and some strains are disproportionately affected. However, the number of transcripts per cell is generally equivalent between strains even when they differ in representation within libraries (*Figure 2—figure supplement 1B*). By contrast, we find that total transcript counts per cell are highly linked to environmental growth conditions (*Figure 2—figure supplement 1C*), which is consistent with decreased total transcriptome pool size in suboptimal conditions (*Athanasiadou et al., 2019*). For cells growing in rich medium (YPD) we recover a median of 2250 unique transcripts per cell, from a median of 695 distinct genes, indicating a capture rate of approximately 3–5% of total transcripts from each cell. The strain genotype does not strongly influence transcript counts per cell (*Figure 2—figure supplement 1D*). There is a high correlation between single-cell expression data and bulk RNA expression data (spearman correlation 0.941) for wild-type cells grown in YPD (*Figure 2—figure supplement 2*) indicating that the effect of technical bias caused by single-cell processing is minimal. We also find good correlation to other published single-cell yeast data sets, and a comparable published bulk RNAseq experiment,.

Mapping the digital expression matrix into two-dimensional space with a Uniform Manifold Approximation and Projection [UMAP] results in clear separation of individual cells into groups based on environmental condition (*Figure 2B*). Cells from different minimal media or nitrogen-limited growth conditions localize near each other, and cells grown in different rich nitrogen sources are clearly separate from each other. Within environmentally-determined grouping there appears to be no strong ordering by genotype (*Figure 2C*). These clusters are not driven by sequencing depth (*Figure 2—figure supplement 3B*), although there are some stress conditions which have subpopulations which are downregulated for ribosomal genes and upregulated for induced environmental stress response (iESR) genes (*Figure 2—figure supplement 3C–F*). The increased relative abundance of ribosomal related gene expression in rich media conditions is consistent with previously-observed correlation of ribosomal gene expression and cellular growth rate (*Brauer et al., 2008*). Some measures of variance per-gene differ in different growth conditions (*Figure 2—figure*

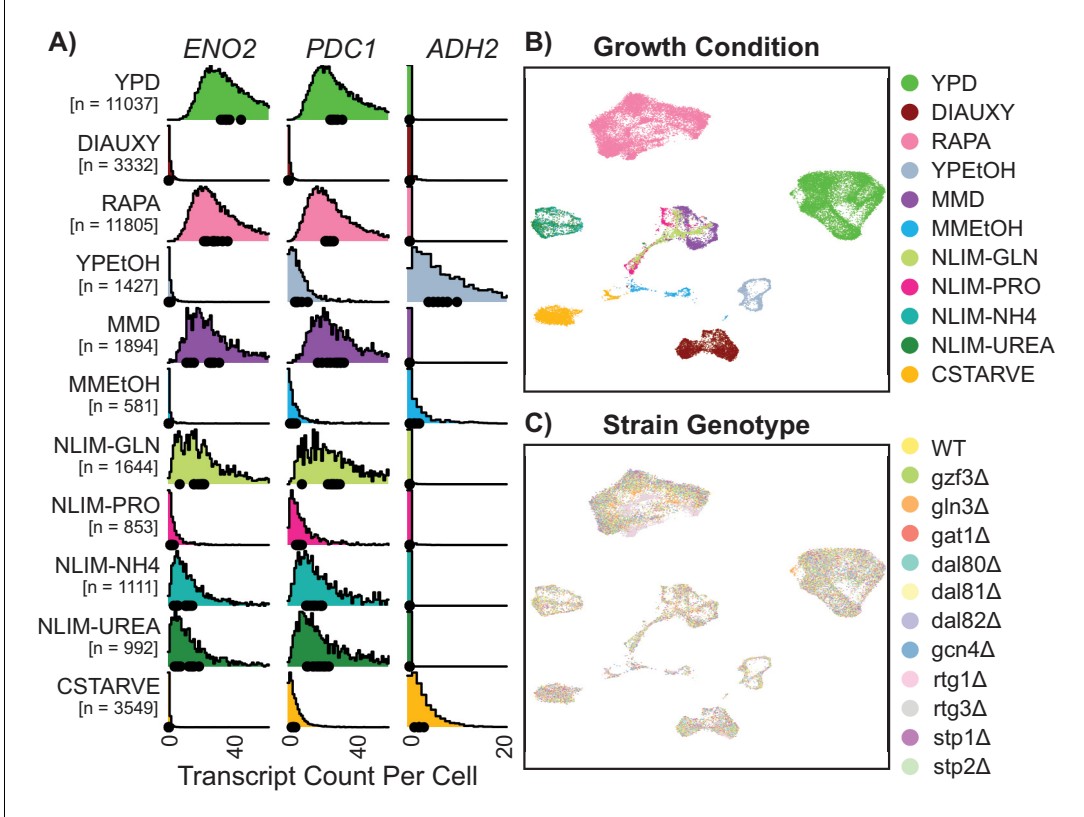

**Figure 2.** Gene expression of single Yeast Cells Cluster Based on Environmental Growth Condition. (**A**) Normalized density histograms of raw UMI counts of the core glycolytic genes *ENO2* and *PDC1*, and the alcohol respiration gene *ADH2* in each environmental growth condition. Mean UMI count for each of the 12 different strain genotypes within each growth condition are plotted as dots on the X axis. (**B–C**) Uniform Manifold Approximation and Projection (UMAP) projection of log-transformed and batch-normalized scRNAseq data. Axes are dimensionless variables V1 and V2 with arbitrary units, here omitted. Individual cells are colored by environmental growth condition (**B**) or by strain genotype (**C**). Growth conditions are abbreviated as in *Table 1*.

The online version of this article includes the following figure supplement(s) for figure 2:

**Figure supplement 1.** Quality Control of Single-Cell RNA Sequencing Data.
**Figure supplement 2.** Single-Cell RNA Expression Comparison.
**Figure supplement 3.** Expression of Categories of Genes in Single Cells.
**Figure supplement 4.** Measures of Gene Variance in Each Condition.

*supplement 4*). Interactive figures are provided (http://shiny.bio.nyu.edu/YeastSingleCell2019/) facilitating exploration of expression levels for all genes.

## The mitotic cell cycle underlies heterogeneity in single cell gene expression

To identify sources of gene expression differences between cells within environments, we clustered single cells within each environmental condition separately by constructing a Shared Nearest Neighbor graph (*Xu and Su, 2015*) and clustering using the Louvain method (*Blondel et al., 2008*). Genes with known roles in mitotic cell cycle are highly represented among the most differentially expressed genes between clusters (*Figure 3—figure supplement 1A*). Overlaying the expression of three of these genes (*PIR1*, *DSE2*, and *HTB1/HTB2*) on UMAP plots illustrates cell cycle effects on single cell gene expression (*Figure 3A* and *Figure 3B*). *PIR1* expression, a marker for early G1 (*Spellman et al., 1998*), is diagnostic of a distinct cluster. *DSE2* is expressed only in daughter cells (*Colman-Lerner et al., 2001*), which allows daughter cells in G1 to be distinguished from mother cells in G1. Cells that have high expression of the histone 2B genes, which are upregulated in S-phase (*Eriksson et al., 2012*), are localized together in the UMAP plots (*Figure 3B*).

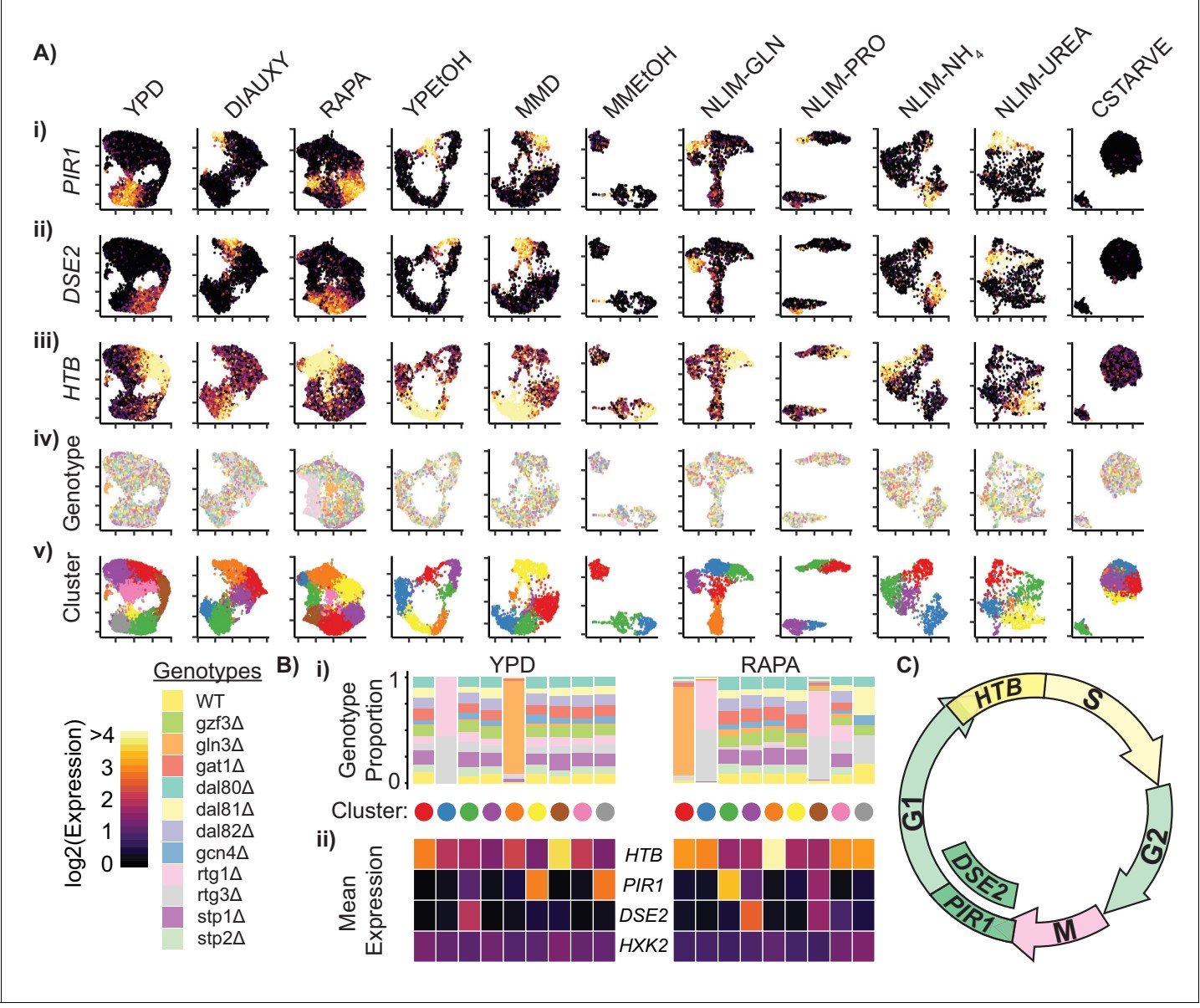

**Figure 3.** Cells Within Conditions Cluster According to Cell Cycle Genes. (A) Cells from each growth condition were separately normalized and transformed into 2-dimensional space using UMAP. The log-transformed, normalized expression for each cell of (i) the G1-phase specific marker *PIR1*, (ii) the G1-phase daughter-cell specific marker *DSE2*, (iii) the S-phase specific marker histone 2B (*HTB*) is shown; (iv) the genotype and (v) the cluster membership of each cell. (B) Summary of clustered single cell expression within the YPD and RAPA growth conditions (i) Proportion of cells from a specific strain genotype within each cluster (ii) The mean log-transformed, normalized expression of the G1- and S-phase marker genes, as well as a hexokinase gene *HXK2* for each cluster (C) Schematic of the mitotic cell cycle with expression of *DSE2*, *PIR1*, and *HTB* genes annotated.
The online version of this article includes the following figure supplement(s) for figure 3:

**Figure supplement 1.** Expression of Important Genes For Clustering.

**Figure supplement 2.** Some Conditions Have Stress Response Clusters Cells from each growth condition were separately normalized and transformed into 2-dimensional space using UMAP.

For each cluster of cells within a growth condition we plotted the proportion of cells belonging to each TF deletion genotype, and the mean expression of several cell cycle genes (*Figure 3B*). Some clusters predominantly contain cells from a single TF deletion genotype; for example, cells deleted for *GLN3* (*gln3Δ*) form a separate cluster in YPD and RAPA conditions, as do cells deleted for one of the RTG heterodimer components (*rtg1Δ* and *rtg3Δ*). However, differences in expression due to

genotype do not appear to be a primary source of expression differences within conditions, as most clusters show a uniform distribution of genotypes (*Figure 3B*, *Figure 3—figure supplement 1B*). Similarly, we do not find that differences in expression of metabolic genes underlie overall differences in expression (e.g. *HXK2*) suggesting that the yeast metabolic cycle (*Silverman et al., 2010*; *Tu et al., 2005*) is not readily identifiable in single cells using scRNAseq. Three of the high-stress growth conditions (NLIM-GLN, NLIM-PRO, and MMEtOH) have clusters that are separate from the majority of the cells analyzed in those conditions. We find that these clusters have higher levels of stress response genes and lower levels of ribosomal genes than other cells in these conditions (*Figure 3—figure supplement 2B–C*) These clusters may reflect cells undergoing early entry into quiescence and provide evidence for a heterogeneous response to stressful conditions.

## Deletion of Transcription Factors causes gene expression changes that differ between growth conditions

To assess our ability to determine differential gene expression between TF knockout strains, we examine the expression of genes known to respond to nitrogen signalling. *GAP1* (General Amino acid Permease) is a transporter responsible for importing amino acids under conditions of nitrogen limitation; *GAP1* expression is regulated by the NCR activators *GAT1* and *GLN3*, the NCR repressors *GZF3* and *DAL80* (*Stanbrough and Magasanik, 1995*), and potentially *GCN4* (*Natarajan et al., 2001*). We identify differing degrees of dysregulation of *GAP1* expression when these TFs are deleted (*Figure 4A*). The effect of deleting TFs varies by condition: *GAP1* is not expressed in YPD and its expression increases in nitrogen-limited media and in response to rapamycin. Deletion of *GAT1* results in decreased expression in nitrogen limiting media, but deletion of *GLN3* does not affect *GAP1* expression. By contrast, in the presence of rapamycin deletion of *GLN3* results in reduced *GAP1* expression. Deletion of *GCN4* only impacts *GAP1* expression in the presence of urea. *MEP2* and *GLN1* are also responsive to nitrogen TFs, and are dysregulated when certain TFs are deleted; expression of the glycolytic gene *HXK2* decreases when *GLN3*, *GCN4*, or *RTG1/RTG3* are deleted, but only in conditions of nitrogen limitation (*Figure 4—figure supplement 1A*). These environmentally dependent impacts of genotype on gene expression demonstrate the importance of exploration of variable conditions for studying genotypic effects on expression.

A variety of statistical methods have been proposed and benchmarked for testing different expression of scRNAseq data (*Soneson and Robinson, 2018*). Our experimental design allows single-cell measurements to be collapsed into a total count (pseudobulk) measurement by summing counts across all cells that correspond to each of the six individual replicates of each genotype within a condition. When we analyze this data using standard approaches to RNAseq analysis (DESeq2) we detect several genes with significant (adjusted p-value<0.05) differences in expression (fold change >1.5) between wild-type and TF deletion strains (*Figure 4B*) that are consistent with known regulatory pathways. There are considerably fewer changes in gene expression as a result of TF deletions compared to the hundreds of genes that change expression between different conditions (*Figure 4—figure supplement 1B*). However, in cells grown in rich media [YPD], we found 96 genes that are differentially expressed in TF deletion strains compared to wildtype (*Figure 4C*), and expression of 160 genes are perturbed in TF deletion strains compared to wildtype when exposed to rapamycin [RAPA] (*Figure 4—figure supplement 1C*). Many of these differentially expressed genes are annotated as functioning in amino acid metabolism and biosynthesis.

## Optimal modeling parameters for network inference from Single-Cell yeast data

Differential gene expression in a TF knockout strain is not sufficient evidence of a direct regulatory relationship as many significant changes in gene expression upon deleting a TF are indirect, and many direct effects may be subtle. Therefore, we constructed a gene regulatory network using the Inferelator, a regression-based network inference method which is based on three main modeling assumptions. First, we assume that Transcription Factor Activity (TFA) is a latent biophysical parameter that represents the effect of a TF binding to DNA and modulating its transcription activity (*Arrieta-Ortiz et al., 2015*; *Fu et al., 2011*). The TFA values are not directly measured, and instead must be estimated as a relative value based on prior knowledge of a regulatory network of TF and target relationships. This TFA estimation is essential as many TFs are post-transcriptionally regulated,

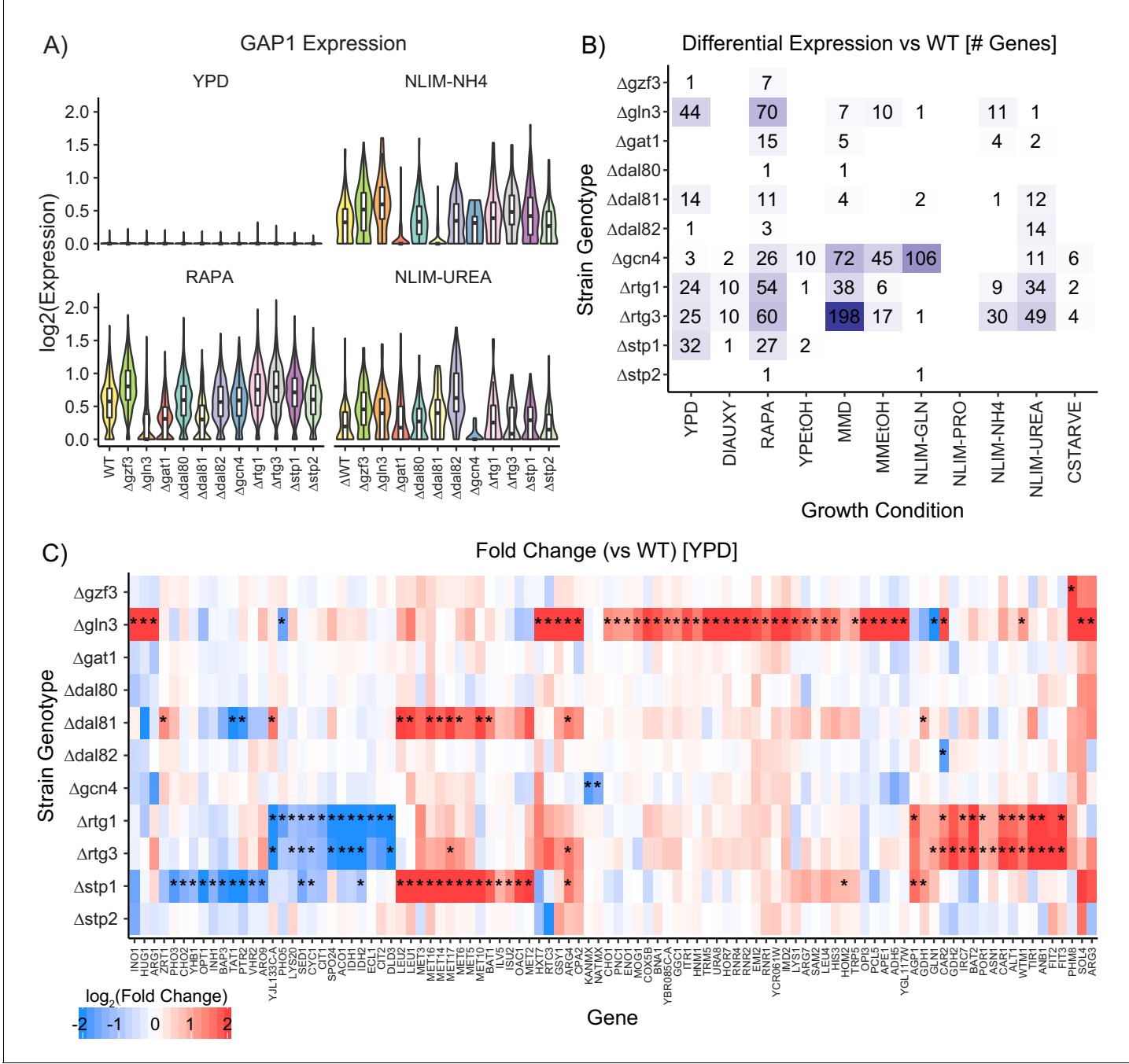

**Figure 4.** Impact of Deleting Transcription Factors on Gene Expression. (A) Violin plots of the log₂ batch-normalized expression of the general amino acid permease gene *GAP1* in YPD, RAPA, ammonium-limited media, and urea-limited media. (B) Count of differentially expressed genes in each combination of growth condition and strain genotype. Data were transformed to pseudobulk values by summing all counts for each the six biological replicates for each genotype and then analyzed for differential gene expression using DESeq2 [1.5-fold change; p.adj <0.05]. (C) Log₂(fold change) of genes differentially expressed in TF knockout strains compared to wildtype, when grown in YPD. Asterisks denote statistically significant differences in gene expression [1.5-fold change; p.adj <0.05].

The online version of this article includes the following figure supplement(s) for figure 4:

**Figure supplement 1.** Differential Gene Expression Varies by Condition.

or are expressed at levels that are not reliably detected by scRNAseq (*Filtz et al., 2014*). Second, we assume that expression of a gene can be described as a weighted sum of the activities of TFs (*Bonneau et al., 2006*) using an additive model in which activators and repressors increase or decrease the expression of targets linearly. Finally, we assume that each gene is regulated by a small number of TFs, and that regularization of gene expression models is required to enforce this biologically relevant property of target regulation. *Saccharomyces cerevisiae*, as a preeminent model organism in systems biology, has a well defined set of known interactions that are of considerably higher quality than is available for more complex eukaryotes providing a validated gold standard for testing model performance (*Tchourine et al., 2018*).

To evaluate the performance of data processing methods and model parameter selections within the Inferelator on scRNAseq data, we perform ten cross-validations using the existing gold standard network. During cross-validation, we infer a GRN using half of the gold standard target genes as priors, then evaluate performance based on recovery of TF-target gene interactions for gold standard interactions that are left out of the priors. We tested preprocessing and prior selection options by inferring networks using gene expression models that are regularized by best subset regression to minimize Bayesian Information Criterion (*Arrieta-Ortiz et al., 2015*; *Greenfield et al., 2013*) and quantified performance in predicting TF-target interactions using the area under the precision-recall curve (AUPR). As negative controls, we employed the same procedure after shuffling priors and after simulating scRNAseq data in which all variance is due to sampling noise. The negative control with shuffled priors establishes a random classifier baseline AUPR of 0.02; the negative control with simulated data establishes a circular recovery baseline AUPR of 0.06 (*Figure 5A*). Performance of the Inferelator on our scRNAseq data far exceeds these baselines, with a mean AUPR of 0.20. This performance from our single dataset is comparable to that of a GRN constructed from 2577 experimental observations using bulk gene expression data (*Tchourine et al., 2018*).

The sparsity of data for each cell acquired using scRNAseq may negatively impact its utility in GRN construction. A commonly used technique to address missing data is data imputation. We tested the impact of several imputation packages on network inference: MAGIC (*van Dijk et al., 2018*), ScImpute (*Li and Li, 2018*), and VIPER (*Chen and Zhou, 2018*). Whereas these methods can enhance separation of gene expression states in low-dimensionality projections (*Figure 5—figure supplement 1A*), we find that they are either ineffective or detrimental to network inference (*Figure 5A*). When the GRN is reconstructed from interactions selected at a precision threshold of 0.5, which takes into account how many interactions are correct according to the gold standard, no imputation method increases the number of recovered interactions compared to unmodified data. Data imputation with MAGIC increases the total number of confidently predicted (confidence >0.95) interactions, but recovers fewer interactions that are correct according to the gold standard.

## Selection of priors for inference from Single-Cell yeast data

Algorithms for network inference perform poorly when making predictions based only on expression data (*Greenfield et al., 2010*). Including prior knowledge of regulatory relationships and network topology improves model selection, and allows approximation of latent variables like TFA. Priors can be generated from regulatory interactions defined using methods such as chromatin immunoprecipitation sequencing (ChIP-seq) or analysis of transposase-accessible chromatin (ATAC-seq) and TF binding motifs, or from curated databases of interactions derived from literature. The source and processing of prior knowledge has a substantial effect on the size and accuracy of the learned network (*Azizi et al., 2018*; *Siahpirani and Roy, 2017*). We tested the impact on GRN reconstruction of prior data derived from literature, and from high-throughput experimental assays that encompass interactions between the entire yeast genome and the majority of known TFs (*Figure 5B*). The best performance is obtained using a curated set of known TF-gene interactions obtained from YEAS-TRACT (*Teixeira et al., 2018*). Generating priors using motif searching within open chromatin regions determined by ATAC-seq (*Castro et al., 2019*; *Miraldi et al., 2019*), and by modeling TF-DNA affinities in promoters (*Ward and Bussemaker, 2008*) provides a considerable improvement over GRN reconstruction from TF expression without priors, but have lower performance than priors derived from curated data.

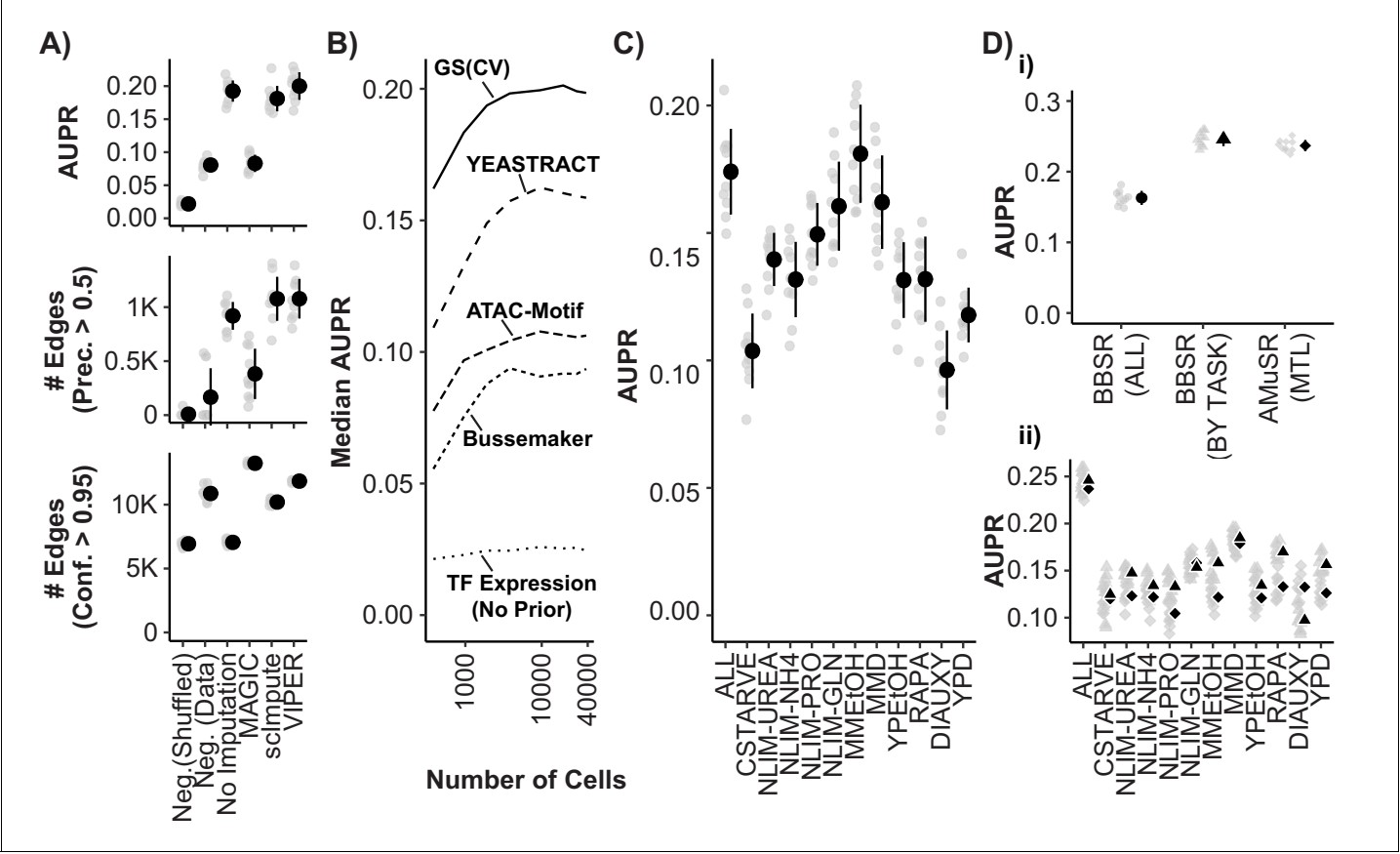

**Figure 5.** Model Performance and Impact of Data Imputation, Prior Selection, and Multitask learning on Network Inference using the Inferelator. (A) Model performance of Inferelator (TFA-BBSR) network inference after shuffling priors [Neg.Shuffled], on a simulated negative data set [Neg. Data], on the unaltered count matrix [No Imputation], and after imputing missing data from the count matrix using the MAGIC, ScImpute, and VIPER packages. Model performance is shown using area under the precision-recall curve [AUPR], as well as the number of network edges using a precision (>0.5) cutoff, and the number of network edges using a confidence (>0.95) cutoff. Each point plotted in gray is a separate cross-validation analysis, with mean +/- one standard deviation plotted in black (n = 10). (B) Median AUPR after cross-validation (n = 10) and resampling to different numbers of cells, for priors extracted from the gold standard [GS], the YEASTRACT database, Bussemaker et al, priors predicted from ATAC-seq data and motif searching, and no prior data. (C) AUPR of separate cross-validation network inference using cells from all growth conditions, or from individual conditions separately. Each cross-validation (n = 10) was downsampled to the same number of cells. (D) Cross-validation (n = 10) using the YEASTRACT prior data. Networks are learned for all conditions together [BBSR (ALL) •], for all conditions individually with TFA-BBSR followed by combination [BBSR (BY TASK) ▲], and for all conditions together in multi-task learning followed by combination [AMuSR (MTL) ◆]. Models are evaluated by (i) AUPR on the aggregate, final network and (ii) AUPR for each task-specific subnetwork from BBSR (BY TASK) (▲) and AMuSR (MTL) (◆).

The online version of this article includes the following figure supplement(s) for figure 5:

**Figure supplement 1.** Low-Dimensional Clustering of Imputed Data Scatter plot after UMAP into 2-dimensional space.

## Multi-task learning improves network inference and enables reconstruction of a unified Gene Regulatory Network from multiple conditions

Numerous methods exist for integrating information across different conditions and experiments that aim to reduce technical variation while retaining biologically meaningful differences (*Hicks et al., 2018*; *Leek et al., 2010*). The appropriate approach to integrating scRNAseq data for the purpose of GRN reconstruction remains unknown. We find that when we separate data based on environmental conditions and infer GRNs we obtain unique networks of differing quality (*Figure 5C*). Learning a single network from all conditions by first combining the data can be compromised by technical variability and imbalance in the number of cells between conditions. Furthermore, normalizing batches to equal transcript depth risks suppressing differences which are true biological variability. An alternative approach is to treat the cells from each environmental condition

as separate tasks. Separate tasks can be learned independently, without sharing information between tasks (implemented as BBSR (BY TASK)). This entails learning networks from each task, and then combining task-specific networks into a global network. Alternatively, networks can be learned together in a multitask learning (MTL) framework (*Lam et al., 2016*), sharing information between tasks while they are learned, which we have implemented as Adaptive Multiple Sparse Regression (AMuSR) (*Castro et al., 2019*). We find that, compared to network inference using all data simultaneously [BBSR (ALL)], treating conditions as separate network inference tasks provides a considerable improvement in performance (*Figure 5Di*). This is likely due to the retention of environmentally specific interactions that would otherwise be obscured using methods for normalizing data prior to GRN construction. The performance of the information sharing network inference approach [AMuSR (MTL)] and the non-sharing network inference approach [BBSR (BY TASK)] are very similar overall. We find that some individual tasks had modest improvements in model performance with AMuSR and others with BBSR (*Figure 5Dii*).

We constructed a global gene regulatory network using the YEASTRACT priors (as determined above) and our multi-task network inference (AMuSR) procedure. Eleven GRNs were jointly learned from each of the eleven environmental growth conditions; for each task a confidence score for each regulator-target interaction was calculated. GRNs learned for each condition were combined by rank summing condition-specific confidence scores to create a global confidence score for each potential interaction. All potential interactions are ranked by global confidence score, and a global GRN is constructed from interactions that meet the precision threshold of 0.5, as measured by recovery of known interactions (*Figure 6A*, *Source code 2*). The resulting GRN comprises 6114 new interactions and 6114 interactions present in the priors, resulting in a total of 12,228 regulator-target interactions. We find that 5372 interactions from the priors are not recovered (recall of 0.532). The global GRN comprises an identified regulator for approximately half of all known genes (*Figure 6—figure supplement 1A*). There is a positive correlation between expression level for a gene and the number of regulators for that gene (*Figure 6—figure supplement 1B*) and 90% of the identified interactions are predicted to have activating effects (*Figure 6—figure supplement 1C*). Many condition-specific networks have uniquely identified interactions (*Figure 6—figure supplement 1D*), but more than 75% of the final network is composed of TF-gene interactions found in multiple conditions (*Figure 6—figure supplement 1E*). Of the novel learned interactions (i.e. those not in the prior data), 60% have evidence of a TF-gene regulatory relationship when compared to the YEASTRACT database (*Figure 6—figure supplement 1F*). 573 learned TF-gene interactions have evidence for physical localization of the TF to the target gene, and 2957 learned TF-gene interactions have evidence of expression changes when the TF is perturbed.

Within the nitrogen-regulated TF subnetwork comprising the 11 deleted TFs (*Figure 6B*) we identify 885 regulator-target interactions, of which 447 are novel, and 438 are present in the priors. This subnetwork contains many features consistent with expectations including co-regulation of targets by the NCR TFs. Overall, the global GRN has the largest number of target genes for general TFs (including *ABF1, RAP1, CBF1,* and *SFP1*), but we also define regulatory relationships for a total of 129 of the predicted 207 yeast TFs (*Figure 6C*). The poorest recovery of prior data is found for TFs that regulate environmental responses not included in our experimental design, such as the stress response TF *MSN2* and the mating TF *STE12*, highlighting the necessity of exploration of condition space for complete network reconstruction. Regulators and target genes can be mapped to Gene Ontology (GO) biological process slim terms, which are broad categorizations that facilitate pathway analysis. Ordering GO slim terms by the number of interactions in the learned GRN, we find that for target genes eight of the top ten GO slim terms are metabolism-related (*Figure 6D* i); in contrast, for regulatory TFs, five of the top ten GO slim terms are stress response related (*Figure 6D* ii).

## Identification of coregulation by cell cycle and environmental response TFs

Analysis of single cell expression in asynchronous cultures allows detection of cell cycle regulated relationships. The learned global GRN contains 257 genes that are regulated both by nitrogen TFs and by cell cycle TFs (*Figure 7A*). Many of these regulatory connections are novel; likely due to the fact that identifying interactions between metabolism and cell cycle are challenging in asynchronous cultures without single-cell techniques. Of these genes, 38 are annotated with the amino acid metabolic biological process GO term and 20 are annotated with the ion or transmembrane transport

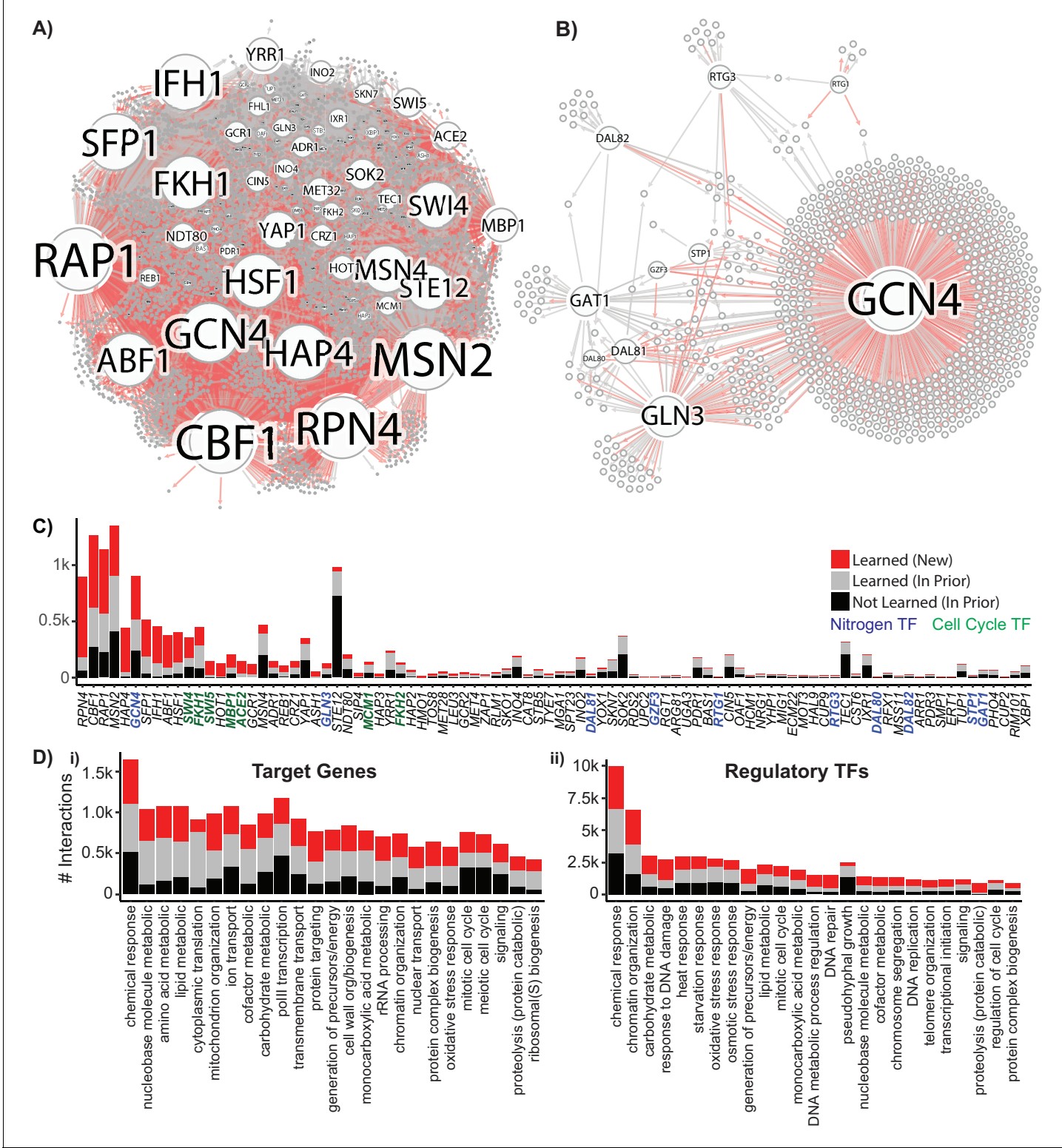

**Figure 6.** Reconstruction of a Gene Regulatory Network Identifies New Regulatory Relationships. A network inferred from the single-cell expression data using multi-task learning and the YEASTRACT TF-gene interaction prior, with a cutoff at precision >0.5. (**A**) Network graph with known interaction edges from the prior in gray and new inferred interaction edges in red (**B**) Network graph of the 11 nitrogen-responsive transcription factors with known edges from the prior in gray and new edges in red (**C**) The number of interactions for each TF; interaction edges present in the prior that are not in the final network are included in black. The nitrogen TFs knocked out in this work are labeled in blue, and TFs with gene ontology annotations for mitotic

*Figure 6 continued on next page*

*Figure 6 continued*

cell cycle are annotated in green (D) Gene ontology classification of network interactions by the GO slim biological process terms annotated for the target gene and the regulatory TF (the GO term <u>transcription from RNA pol II</u> is omitted from the annotations for regulatory TFs).

The online version of this article includes the following figure supplement(s) for figure 6:

**Figure supplement 1.** Summary of Learned GRN.

biological process GO term. Only 11 are annotated with the mitotic cell cycle biological process GO term, indicating that the majority of the interconnection between cell cycle and nitrogen response genes is due to regulation of metabolism-related genes by cell cycle TFs.

We estimated the TFA for every TF in each cell, using the learned GRN and the single-cell expression matrix. The TFA of nitrogen responsive TFs is principally linked to growth condition as these TFs vary in activity between conditions (*Figure 7A*), but are generally similar within condition (*Figure 7—figure supplement 1A*). As expected, we find that cells grown in rich media (YPD) have low TFA for the NCR TFs *GLN3* and the GAAC TF *GCN4*. The TFA for these TFs increases substantially upon treatment with rapamycin. By contrast, the estimated TFAs of cell cycle TFs varies within condition (*Figure 7B*); and are concordant with cell cycle responsive gene expression (*Figure 7—figure supplement 1B–D*).

## Discussion

### A robust scRNAseq and transcriptional barcoding method in yeast

Since the inception of single-cell RNA sequencing (*Tang et al., 2009*), technological advances have resulted in the scale of datasets increasing from tens of cells to tens of thousands in a diversity of organisms. However, the number of cells recovered during scRNAseq in budding yeast has been comparatively limited in studies published to date (*Gasch et al., 2017*; *Nadal-Ribelles et al., 2019*). We present here the first report of droplet-based scRNAseq in this widely used model eukaryotic cell. Using a diverse library of transcriptionally barcoded gene deletion strains we were able to efficiently analyze the gene expression state of 38,255 cells using 11 experiments. In addition to facilitating multiplexed analysis of genotypes, transcriptional barcoding provides a facile means of identifying doublet cells within droplets thereby increasing the accuracy of single cell analysis.

Consistent with our understanding of global gene expression variation first characterized in foundational studies of the transcriptome (*DeRisi et al., 1997*; *Gasch et al., 2000*), we find that environmental condition is the primary determinant of the gene expression state of individual yeast cells. However, we observe significant heterogeneity in individual cell gene expression within conditions. Much of this variation can be explained by the mitotic cell-cycle. It is important to note that we do not remove or suppress this cell-cycle driven variance. The cell cycle is itself driven by transcriptional regulators, and our goal is to build a network that integrates cell-cycle regulation with regulated responses to the environment. The ability to access the crosstalk between signalling pathways and the cell cycle program is a key advantage to performing single-cell sequencing in asynchronous cultures, which bypasses many of the limitations of synchronized bulk sequencing experiments. It is also important to note that in several stressful growth conditions, we see heterogeneous cellular responses; some cells appear to be proliferative, while other cells have downregulated translational machinery and upregulated stress response genes. This is an interesting outcome by itself, as it is further evidence of bet-hedging strategies (*Levy et al., 2012*), and we expect that the presence of multiple distinct transcriptional states between cells in the same environmental condition is advantageous for network inference. Model performance, as measured by AUPR, can vary considerably when learning networks from any single growth condition (*Figure 5C*). Cells in rich YPD media do not require many anabolic pathways to be active, and primarily express genes required for the cell-cycle, translation, and glycolysis; in contrast, cells in minimal MMD media must express these pathways plus many anabolic pathways to synthesize nitrogenous bases, cofactors and amino acids. We find that this increased transcriptional diversity results in better overall performance. Nonetheless, the largest performance gain comes from aggregating networks from cells in different conditions (*Figure 5D*), which demonstrates a general advantage to learning GRNs from heterogeneous data.

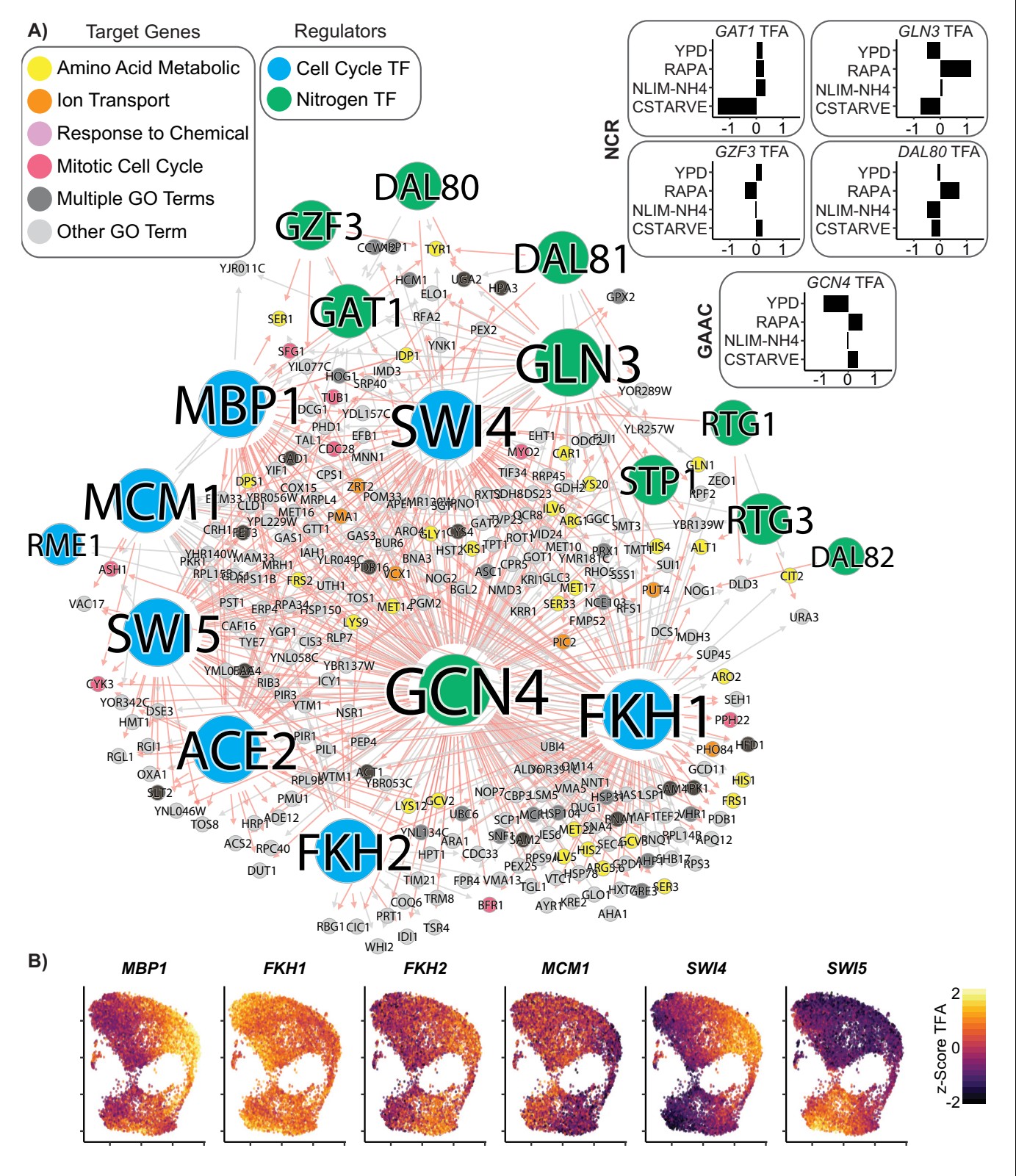

**Figure 7.** Coordinated regulation of Nitrogen Response and Cell Cycle. (A) A gene regulatory network showing target genes that are regulated by at least one nitrogen TF (blue) and at least one cell cycle TF (green). Target gene nodes are colored by GO slim term. Newly inferred regulatory edges are red and known regulatory edges from the prior are in gray. Transcription factor activity (TFA) is calculated from the learned network and then scaled to

*Figure 7 continued on next page*

*Figure 7 continued*

a z-score over all cells which do not have that TF deleted (e.g. *gcn4Δ* cells are omitted from the calculation for *GCN4* TFA). The mean TFA z-score for four selected conditions is inset for GAAC and NCR TFs (**B**) TFA for cell cycle TFs for each cell in the YPD growth condition.

The online version of this article includes the following figure supplement(s) for figure 7:

**Figure supplement 1.** Cell Cycle TF Activity Clusters within Growth Conditions.

Deletion of specific transcription factors results in changes in single cell gene expression for some TFs in some conditions. However, genotypic effects are comparatively minor. We believe that this is due to multiple factors including functional redundancy between TFs, physiological adaptation to the genetic perturbation and the conditional specificity of TFs. It is likely that perturbations that are transiently induced, and result in increased TF activity (*McIsaac et al., 2013*) may be effective in eliciting detectable responses in gene expression, facilitating causal inference. The use of precise gene deletions does provide several advantages over the use of CRISPR/Cas9-based perturbations as engineered deletions are unambiguous whereas the efficiency of perturbation by CRISPR/Cas9 varies for different guide RNAs.

## A generalizable framework for GRN construction using scRNAseq

Constructing GRNs from single cell gene expression data is a universal goal in all organisms. A yeast single-cell expression matrix has several beneficial properties for design and testing of gene regulatory network inference models as there exist high quality known interactions and TF binding motifs. The issues of data sparsity and low sampling rates are likely to be problems common in experiments in any organism using scRNAseq. We find that techniques that have been developed for normalization and imputation do not improve performance of the additive linear model-based inference of the Inferelator algorithm (*Figure 5*). However, there are significant opportunities for development of smoothing techniques that would enhance network inference, perhaps targeting latent biophysical parameters like transcription factor activity. It seems reasonable to assume that these biophysical parameters should be stable within the local neighborhood of samples, and the activity calculation that we have used is ill-conditioned and potentially unstable. This is of particular concern when working with undersampled single-cell data and we are actively addressing this issue.

We find that the application of multitask learning is well suited to GRN reconstruction from scRNAseq data. Jointly learning multiple related tasks improves generalization accuracy, especially in scenarios in which datasets are undersampled (*Caruana, 1998*), and has the desirable side benefit of mitigating the need for complex batch-correction techniques that aim to address technical variation between experiments. Removing batch-effect technical noise from data without suppressing interbatch biological variability remains an unsolved problem, and therefore application of multitask learning approaches to network inference from single-cell data is likely to be generally applicable to integrating scRNAseq data from different cell types and conditions.

## A global GRN for budding yeast

Using our scRNAseq dataset, we reconstructed a global GRN with several novel regulatory relationships. Among the most novel of these interactions are those between cell-cycle associated TFs and targets and nitrogen TFs and target genes. The cell cycle and metabolism are, by necessity, interconnected, and the mechanism of rapamycin in arresting cell cycle through TOR is well-established (*Heitman et al., 1991*). Several studies have identified metabolic cycling patterns which are believed to be driven by the cell cycle (*Burnetti et al., 2016*; *Slavov and Botstein, 2011*; *Tu et al., 2005*). Although regulatory connections between environmental sensors, metabolism, and the cell cycle have been previously reported, a comprehensive regulatory network does not exist, in large part because of the difficulty of experimentally perturbing cell cycle without confounding metabolic changes. Our study provides a valuable first step in identifying specific regulatory connections that were previously inaccessible, and which are necessary to create a complete map of the yeast regulome.

Incorporation of additional information into the network inference process, including information about interactions between transcription factors such as functional redundancy and heterodimerization, would likely improve learning of the network. We note that several TFs have few learned

targets reflecting the requirement for surveying conditions in which particular TFs are active. For example, *STE12* and *TEC1* are mating-related TFs that we expect to be entirely inactive in our diploid cells; *MSN2* and *YAP1* are stress-responsive TFs that respond to specific stimuli that were not tested in our study. Targeted analysis of the GRN with rationally designed genetic perturbations and environmental conditions will maximize the additional information that can be recovered from future experiments.

## Conclusion

Single-cell sequencing is a transformative method for systems biology. To date, scRNAseq has been widely applied to the problem of defining different cell types. However, the ability to simultaneously study the expression of hundreds of genotypes in different conditions, and sample the expression state of thousands of cells, presents a rich source of information for the purpose of GRN reconstruction. Our study implements this approach in budding yeast, the workhorse of systems biology, and establishes a generalizable framework for GRN reconstruction from scRNAseq data in any organism.

# Materials and methods

Requests for strains and reagents should be directed to David Gresham (dgresham@nyu.edu). Requests related to computational analysis and code should be directed to Richard Bonneau (rb133@nyu.edu). There are no restrictions on the materials or the code used in this work. All materials are released under CC-BY 4.0 and all code is available under the permissive MIT or BSD licenses.

## Yeast strain construction and growth

All yeast strains were generated from the prototrophic FY4 (MAT**a**) or FY5 (MAT**α**) background strains. Yeast were transformed using the standard lithium acetate transformation protocol (*Gietz and Schiestl, 2007*). *E. coli* were transformed using the standard chemically competent transformation protocol. Plasmid constructions were confirmed by sanger sequencing. Yeast genotypes, plasmid sequences, and oligonucleotide sequences are provided as *Supplementary file 1*-supplemental tables 1-3. Media formulations are provided as *Supplementary file 1*-supplemental table 4.

### Construction of barcoded deletion cassettes

The deletion cassette plasmid was constructed by amplifying pTEF::KAN$^R$ from pUG6 (Euroscarf) and tTEF from pUG6, with an overlapping junction between KAN$^R$ and tTEF containing two BbsI sites for golden-gate mediated barcode cloning. These pieces were assembled into pUC19 using gibson isothermal assembly to generate DGP304. This plasmid was then modified by linearizing with BamHI and XbaI, amplifying a bacterial GFP expression cassette from pWS158 (Addgene), and assembled using gibson isothermal assembly to generate DGP306.

Gene deletion barcodes were created by synthesizing an oligonucleotide containing flanking PCR handles (M13F and M13R), flanking BbsI sites for golden gate cloning, and the degenerate sequence caNNgNNgtNNgNNgtNNgNNgt. The mixture of oligonucleotides was double-stranded using *E. coli* DNA Polymerase I, Large (Klenow) fragment. Klenow buffer (1x NEB Buffer 2.1 [NEB #B7202S]) was mixed with 250 nM barcode oligonucleotide, 250 nM M13R primer, 200 nM/each dNTP [NEB #N0447S], incubated at 80℃ and slowly cooled to room temperature. The DNA Polymerase I, Large (Klenow) Fragment (NEB #M0210S) was added to 0.1 U/µL and the reaction was incubated at 37℃ for 30 min. The polymerase was heat-inactivated by placing the reaction at 75℃ for 20 min. The resulting dsDNA cassette was used with no further cleanup.

The barcode was inserted into the 3′ untranslated region of the pTEF::KAN$^R$::tTEF yeast selection marker cassette in DGP306 by BbsI-mediated golden gate cloning. A golden gate reaction was prepared with 1x Thermo FastDigest Buffer [Thermo #ER1011], 1 mM ATP, 10 mM DTT, 2 U/µL T4 DNA Ligase [NEB #M0202S], 1 U/µL BpiI [Thermo #ER1011], 10 ng/µL DGP306, 25 nM barcode dsDNA, and incubated in a thermocycler using the following program: 37℃ 20 min; 25x cycles of 37℃ for 5 min and 16℃ for 5 min; 37℃ for 20 min; 80℃ for 20 min. An additional 1 U/µL BpiI was then added to the reaction mix and incubated at 37℃ for 30 min to linearize any remaining uncloned plasmid.

The golden gate cloning reaction was transformed into One Shot TOP10 *E. coli* (ThermoFisher #C404003). Cloning and transformation efficiency was estimated by plating 2% of the transformation

onto LB + ampicillin plate and counting GFP$^+$ and GFP$^-$ colonies. The remainder of the reaction was inoculated into 200 mL molten LB + ampicillin + 0.6% (w/v) SeaPrep Agarose (Lonza 50302) media, thoroughly mixed, snap cooled in an ice bucket, and incubated overnight at 37°C. The soft agar culture was then collected by centrifugation, washed with PBS, and resuspended in 2 mL 50% glycerol. 100 µL of this mixture was used to inoculate a culture of 100 mL LB + ampicillin and the remainder stored at −80°C in aliquots. The 100 mL culture was grown for 8 hr at 37°C, harvested, washed with PBS, and stored at −20°C until midiprepped (Qiagen) according to the manufacturer's protocol.

## Construction of a barcoded Transcription Factor deletion array

The degenerate barcoded plasmid was used as template for PCR using primers containing gene-specific targeting homology arms (1x NEB Q5 Master Mix #M0494S, 1 ng template plasmid, 250 nM/each oligo). The PCR amplicon was then transformed into FY4 and plated on YPD+G418 to select transformants. Transformants containing the gene deletion were confirmed using colony PCR and gene-specific primers and a KANR primer. PCR products of correct transformants were cleaned using silica spin columns (Qiagen) according to the manufacturer's protocol and the barcode identified by Sanger sequencing. At least six uniquely barcoded strains (i.e. biological replicates) were generated for each genotype, with the criteria that each barcode had to differ by at least three bases, ensuring that the probability of barcode collisions is extremely low.

The plasmid DGP328 (pTEF::NATR::tTEF) was used as template for PCR using primers containing the same gene-specific targeting homology. The PCR amplicon was transformed into FY5 and plated on YPD+nourseothricin. Positive transformants were confirmed using colony PCR with gene-specific primers and a NATR primer.

FY4-derived MATa strains were arrayed in a 96-well plate (Corning 3788) and then pinned (V and P Scientific #VP407FP12) onto YPD in an OmniTray (Nunc 165218). FY5-derived MATα strains were arrayed in a 96-well plate so that the same gene was disrupted in matching wells of the MATa and MATα plates and then pinned onto YPD. These arrays are grown overnight at 30°C. The MATa array and MATα array were then pinned to the same YPD plate to create spots where MATa and MATα strains were overlaid. The plate layout was designed so that some locations had only MATa strains, only MATα strains, or no strains, to control for mating, contamination, and the efficacy of diploid selection. The mating array was grown overnight at 30°C to allow mating to occur and then pinned to a YPD+G418+nourseothricin plate to select for MATa/MATα diploids. This diploid selection plate was grown overnight and then pinned to a YPD+G418+nourseothricin plate for a second round of diploid selection. The second diploid selection plate was grown overnight at 30°C and then pinned to a YPD+G418+nourseothricin plate for a third round of diploid selection at 30°C. This plate was then pinned to several replicate 96 well round-bottom plates containing 200 µL YPD+G418+nourrseothricin in each well. These plates were cultured with shaking overnight at 30°C, then centrifuged and the media aspirated. The cells were resuspended in 50% glycerol and the plates stored at −80°C.

## Culturing and harvest

The barcoded deletion array was pinned from a frozen stock plate at −80°C onto a YPD plate for recovery and grown overnight at 30°C. The first recovery plate was then pinned to a second recovery YPD plate and grown overnight at 30°C. The second recovery plate was pinned to a 96 well round-bottom plate containing 200 µL YPD in each well and grown overnight at 30°C. The cultures from this plate were pooled, washed 2x with 50 mL PBS, and then resuspended in 1 mL PBS. 250 µL of the washed cells were used to inoculate 50 mL of the relevant media for the specific experimental condition in a shake flask. These flasks were grown for 4 hr. The experiment grown to diauxic shift was grown for 10 hr. We confirmed that glucose in the media was exhausted between hour 9 and hour 10 using a hexokinase-based assay (R-Biopharm #10716251035). All other steps of harvesting cells were identical to the 4 hr experiments. The experiment treated with rapamycin was grown for 3 hr and 30 min in YPD, and then 10 µL of rapamycin stock (1 mg/mL Millipore #553210 in ethanol) was added to a final concentration of 200 ng/mL. Cells were then harvested at 4 hr (after 30 min in rapamycin).

Cell count per mL at harvest was determined using a Beckman Coulter Z2 Particle Counter #6605700. Cell density (cells/mL) for each condition at harvest was as follows: (YPD 1.4e7; RAPA

1.2e7; YPEtOH 1.0e7; NLIM-GLN 0.5e7; NLIM-NH4 0.8e7; NLIM-PRO 0.4e7; NLIM-UREA 0.5e7; MMD 1.1e7; MMEtOH 0.7e7; CSTARVE 0.1e7) A volume of culture containing ~$10^8$ cells was collected and the cells pelleted by centrifugation. These cells were immediately resuspended in 1 mL RNALater (Qiagen #76104), washed 2x with 1 mL RNALater and resuspended in a final volume of 500 µL RNALater. This suspension was stored at −20°C for 12 to 72 hr.

## Library preparation and sequencing
All steps below used RNAse-free reagents.

### Single cell library preparation
Cells stored in RNALater were removed from −20°C and ~$10^7$ cells were washed 2x with 1 mL spheroplasting buffer (50 mM Sodium Phosphate pH 7.5, 1M Sorbitol, 10 mM EDTA, 2 mM DTT, 100 µg/mL BSA). Cells from fermentative phase growth cultures were then resuspended in 100 µL spheroplasting buffer + 0.1 U Zymolyase 100T (Zymo Research #E1004). Cells from respiratory phase growth cultures or starvation cultures were resuspended in 100 µL spheroplasting buffer + 0.25U Zymolyase 100T. The spheroplasting reaction was incubated at 37°C for exactly 20 min, and then the spheroplasted cells were pelleted and resuspended in 500 µL RNALater for 5 min on ice. After this incubation the spheroplasted cells were pelleted and washed 3x with 1 mL wash buffer (10 mM TRIS pH 8, 1M Sorbitol, 100 µg/mL BSA) and resuspended in 1 mL wash buffer. The cells were visualized to confirm spheroplasting and counted using a hemocytometer. A dilution equal to ~$5 \times 10^6$ cells/mL in wash buffer was prepared and then immediately used for single cell isolation.

Single cell library preparation was done using the 10x Genomics Chromium 3' v2 Single Cell Gene Expression Kit (10x Genomics #120237), following the kit protocol. 66.2 µL of single-cell master mix was prepared to which 27.7 µL $H_2O$ was added. The microfluidic Chromium Single Cell A Chip (10x Genomics #120236) was then prepared for use. 6 µL of prepared spheroplast cell suspension was added to the single-cell master mix, and then immediately transferred to the microfluidics chip. Hydrogel beads and partitioning oil were added according to the manufacturer's protocol, and the cells were encapsulated with hydrogel beads using the 10x Genomics Chromium Controller. Following emulsification, reverse transcription and cleanup was performed according to the manufacturer's protocol. Whole transcriptome amplification was performed using a total of 10 cycles of PCR. Cleanup, fragmentation, adapter ligation, and indexing was performed according to the manufacturer's protocol, using 8–10 cycles of PCR for the indexing reaction.

Transcribed barcodes were amplified from the whole transcriptome amplification prior to fragmentation. The KAN$^R$ transcript containing the genotype barcode was amplified in a reaction(1x KAPA HiFi Hotstart Readymix [Kapa #KK2602], 200 nM/each primer, 1 µL 10x whole-transcriptome DNA), using 6 cycles of PCR (98°C for 3:00; 6 cycles of 98°C for 0:20, 63°C for 0:20, and 72°C for 0:20 min; 72°C for 1:00 min). The amplicon pool was then purified with 1x volume of SPRIselect beads (Beckman Coulter #B23317) and eluted into 24 µL $H_2O$. To this eluate, 25 µL of 2x KAPA HiFi Hotstart Readymix was added, as well as 200 nM/each indexing primers. The indexing reaction was cycled for 8–10 cycles of PCR, using the 10x Genomics indexing PCR reaction settings (98°C for 0:45; 8-10x cycles of 98°C for 0:20, 54°C for 0:30, and 72°C for 0:20; 72°C for 1:00).

Library fragment sizes were determined using a High Sensitivity D1000 Screentape (Agilent #5067–5584) and quantified with the KAPA illumina library quantification system (Roche #KK4953) on a Roche lightcycler 480. Libraries from each condition were pooled so that 99% of the pool consisted of the single-cell transcriptome library and 1% of the pool consisted of the genotype barcode amplicon. Samples were then pooled for multiplex sequencing on an Illumina NextSeq 500 with the NextSeq 500/550 v2.5 High Output 150 Cycle kit (Illumina #20024907), using the sequencing parameters recommended by 10x Genomics (Read 1: 26 bp, Read 2: 98 bp, Index 2: 8 bp) and standard illumina read and indexing primers.

### Bulk RNA library preparation
Each of the six wild-type yeast strains (MAT a/α Δho::KanMX/Δho::NatMX) were separately grown overnight in YPD at 30°C.~$10^8$ cells (0.5 mL) of overnight culture was subcultured into separate 50 mL flasks of pre-warmed YPD and cultured with shaking for 4 hr at 30°C. At 4 hr, for each culture

flask,~$10^8$ cells were pelleted by centrifugation and immediately transferred to a microfuge tube, then snap-frozen in liquid nitrogen for storage at −80℃.

For each of six wild-type samples snap-frozen in liquid nitrogen and stored at −80℃, cell pellets were removed from −80℃ storage and immediately resuspended in 1 mL TRIZOL (ThermoFisher #15596026), which is an organic extraction reagent with phenol and the chaotropic salt guanidinium thiocyanate (*Chomczynski and Sacchi, 2006*). After sitting at RT for 5 min, 200 µL chloroform was added and tubes were mixed by inversion. Organic and aqueous phases were separated by centrifugation at 4℃. The aqueous phase was re-extracted with 500 µL acid phenol:chloroform (ThermoFisher #AM9720), then the aqueous phase from that extraction was re-extracted with 500 µL chloroform. 1:10th volume 5M NH$_4$OAc (ThermoFisher #AM9070G) and 2.5x volumes of ice-cold absolute ethanol were added to the aqueous phase from the chloroform extraction, and RNA was precipitated overnight at −80℃. After precipitation, the RNA pellet was washed with ice-cold 70% ethanol and dissolved into 100 µL RNA elution buffer (10 mM TRIS pH8, 0.05% TWEEN-20). RNA was quantified by Qbit (ThermoFisher #Q10210) and a working stock of 5 ng/µL RNA was prepared for each sample.

15 µL of reverse transcription mix (5 µL 5x Maxima RT Buffer, 5 µL 20% (w/v) Ficoll PM-400 [GE Life Sciences #17030010], 2.5 µL 10 mM/each dNTP [New England Biolabs #N0447S], 0.5 µL Lucigen NxGen RNase Inhibitor [Lucigen #30281–1], 0.5 µL 50 µM Template Switch Oligo [IDT], 0.5 µL 50 µM Barcode/UMI/poly-dT Oligo [IDT], 0.5 µL Maxima H Minus Reverse Transcriptase [ThermoFisher #EP0752], 0.5 µL H$_2$O) was added to 50 ng (10 µL) of RNA. Each reaction contained a separate barcoded poly-dT oligo such that each of the six biological replicate samples contain a unique, identifiable barcode sequence. Reverse transcription was carried out at 53℃ for 1 hr, followed by heat inactivation at 85℃ for 5 min. 98 µL RLT Buffer [Qiagen] and 2 µL MyOne Silane beads [ThermoFisher #37002D] were added, mixed, and allowed to sit at RT for 10 min. cDNA was then isolated by magnetic separation of beads, followed by 2x washes with 200 µL 80% ethanol. Beads were pooled together and all cDNA was eluted into 40 µL of DNA elution buffer (10 mM TRIS pH8, 0.05% TWEEN-20, 1 mM DTT). 60 µL WTA master mix (50 µL 2x KAPA HiFi Hotstart Readymix, 1 µL 100 µM Forward Oligo, 1 µL 100 µM Reverse Oligo, 8 µL H$_2$O) was added and whole transcriptomes were amplified using 12 cycles of PCR (98℃ for 3:00; 12 cycles of 98℃ for 0:20, 55℃ for 0:20, and 72℃ for 1:15 min; 72℃ for 3:00 min). The amplified pool was then purified with 0.6x volume of SPRI-select beads and eluted into 25 µL DNA elution buffer. Amplified DNA was quantified using a high sensitivity D5000 ScreenTape (Agilent #5067–5592).

Amplified whole-transcriptome DNA was tagmented with a nextera XT kit (Illumina #FC-131–1096) as follows. 3 ng of DNA was diluted to a total volume of 10 µL with DNA elution buffer. 20 µL TD buffer and 10 µL ATM was added and DNA was tagemented at 55C for 10 min. The reaction was halted with 10 µL NT buffer, and the fragment pool was indexed by adding 30 µL NPM buffer, 5 µL illumina index 2 (i7) adapter primer, 5 µL 5 µM DG1954 (no-index primer) and amplifying using 12 cycles of PCR (95℃ for 0:30; 12 cycles of 95℃ for 0:10, 55℃ for 0:30, and 72℃ for 0:30 min; 72℃ for 5:00 min). Libraries were purified by double-sided SPRI selection. 55 µL SPRIselect beads (0.55x) were added to the nextera indexing reaction, and the unbound supernatant was transferred to a clean tube. 20 µL SPRIselect beads were added (0.75x total), and after binding and washing, DNA was eluted into 20 µL DNA elution buffer. Libraries were checked for size with a High Sensitivity D1000 Screentape, and quantified with the KAPA illumina library quantification system on a Roche lightcycler 480. Libraries were sequenced on an Illumina NextSeq 500 with the NextSeq 500/550 v2.5 High Output 150 Cycle kit, using the sequencing parameters recommended by 10x Genomics (Read 1: 26 bp, Read 2: 98 bp, Index 2: 8 bp) and standard illumina read and indexing primers.

## Processing sequencing data

Sequencing results were analyzed using the Cellranger pipeline (10x Genomics) v2.1.0 and custom python scripts written for this project, which are located in the fastqTomat0 GitHub repository (https://github.com/flatironinstitute/fastqToMat0). The reference genome was obtained from Ensembl (Version R64-1-1) as a FASTA file, and the reference annotations were obtained from Ensembl (Version R64-1-1.93) in GTF format. The reference transcript annotations were altered to incorporate 5' and 3' untranslated regions using data from generated using TIF-seq (*Pelechano et al., 2013*) and the gffAnnotate.py script from fastqTomat0. The antibiotic resistance

marker cassettes was added to both the FASTA and GTF files using command line tools. A STAR reference genome was then created from the modified GTF and FASTA files using cellranger mkref.

Raw single-cell sequencing reads were converted into FASTQ files using cellranger mkfastq and a 10x Genomics Index CSV file. These FASTQ reads were then aligned to the reference genome and counted using cellranger count. The FASTQ files for indexes not corresponding to the 10x single-cell transcriptome library were processed with the fastqBCLinker.py script from fastqTomat0, which identifies the genotype for each single-cell read and creates a TSV file mapping cell barcodes to genotypes. The count data from cellranger count and the barcode data from fastqBCLinker.py was combined by the tenXtomatrix.py script from fastqTomat0. This processing step discards doublet single-cell reads, which are identified by removing 'cells' which map to more than one of the 72 genotype-specific barcodes. We expect that 1/72 of these doublets will have the same barcode, and so we expect that ~ 98.5% of doublets will be removed and ~1.5% will be retained. This script produces a dense TSV matrix of counts per gene per cell that can be imported with python's `pandas.read_table()` or R's `read.table()`. This matrix is provided as *Source code 2*. This final data matrix is assembled from 11 independent single-cell sequencing batches, each corresponding to a single shake flask with a different growth condition.

Raw bulk RNA sequencing reads were converted into FASTQ files using bcl2fastq. These FASTQ reads were then aligned to the reference genome and counted using cellranger count, after adding the appropriate custom chemistry configuration and barcode whitelist to cellranger. The count data from cellranger count was processed by the tenXtomatrix.py script from fastqTomat0 into a TSV matrix of counts per gene per sample, which is included in *Source code 1*.

## Network inference

### Inferelator

Network inference with the Inferelator consists of three major steps; data preprocessing and filtering, estimation of transcription factor activities, and regularized regression. Cross-validation of network inference parameters was performed by randomly selecting half of the genes in the gold standard network and removing them. To prevent circularity, any genes that were used in the gold standard were removed from the prior data during cross-validation; for tests where the gold standard network was also used as a prior, this meant that half of the genes in the gold standard network were retained and defined as the gold standard, and half of the genes in the gold standard network were used as priors. A summary table of the cross-validation results is provided as *Supplementary file 1*-Supplemental Table 5.

The randomized negative control was performed by randomly reassigning gene names in the prior data. Transcription factor labels and expression values were otherwise unchanged. The simulated negative control was performed using simulated data by constructing a probability distribution for the yeast transcriptome from estimates of absolute mRNA abundances (*Lahtvee et al., 2017*) and randomly sampling this distribution using the synthesize_data.py script from the fastqToMat0 package. Metadata and total UMI count for each cell were retained in this negative control; only the individual gene counts were synthesized from the simulated control probability distribution.

See below for details on each step of the network inference procedure.

### Single-Cell preprocessing and filtering

Single cell data was loaded as an integer UMI count matrix (Cells x Genes). Genes with a variance of 0 for all cells were removed. The count matrix was then transformed by log scaling using $\log_2(x+1)$. For data sets that had already undergone library normalization and transformation as a result of an imputation method, this transformation preprocessing step was skipped.

### Single-Cell imputation

All imputation methods used the untransformed integer UMI count matrix (Cells x Genes) in which genes with a variance of 0 had been removed. For MAGIC, count data was library size normalized with the `library.size.normalize()` function from the Rmagic package, then transformed by square-root, and subjected to imputation with the `magic()` function from the Rmagic package. For VIPER, count data was subjected directly to the `VIPER()` function from the VIPER package, using the parameters recommended by the VIPER authors for 10x genomics UMI count data. For

ScImpute, count data was normalized by the method included in the ScImpute package and then subjected to imputation with the `imputation_wlabel_model8()` function from the ScImpute package. The R script to perform these imputations is included with *Source code 1*.

## Construction of known prior TF-Gene networks

Construction of the gold standard prior network has been previously described (*Tchourine et al., 2018*); this gold standard network consists of 1403 signed [−1, 0, 1] interactions, for which sign represents activation (+) or inhibition (-), in a 998 genes by 98 transcription factors regulatory matrix. YEASTRACT priors were retrieved from the YEASTRACT database (*Teixeira et al., 2018*) using the *generate regulation matrix* tool. Both activation and inhibition interactions were included, but only those that are supported by both DNA binding and expression evidence. The YEASTRACT prior network consists of 11486 unsigned [0, 1] interactions in a 3912 genes by 152 transcription factors regulatory matrix. Construction of the ATAC-motif priors has been previously described (*Castro et al., 2019*; *Miraldi et al., 2019*), and are built from chromatin accessibility data and known transcription factor binding motifs. The ATAC-motif prior network consists of 71,865 signed integer interactions with a range of [−11,. .., 26], for which sign represents activation (+) or inhibition (-) and absolute values represent the number of motif occurrences, in a 5551 genes by 138 transcription factors regulatory matrix. Bussemaker-priors were generated from modeling transcription factor affinities for regulatory DNA motifs (*Ward and Bussemaker, 2008*). The Bussemaker prior network consists of unsigned floating-point values [0, 20] that reflect estimated binding affinities in a dense 6516 genes by 123 transcription factors regulatory matrix.

## Estimating transcription factor activities (TFA)

Log-transformed single-cell data was transposed into matrix $X$, in which columns are individual cells and rows are genes. $P$ is the connectivity matrix of known prior regulatory interactions between transcription factors (in columns) and genes (in rows). $P_{i,k}$ is zero if there is no known regulatory interaction between transcription factor $k$ and gene $i$. $A$ is the activity matrix, where the columns are the individual cells as in $X$ and rows are the transcription factors. We model the expression of gene $i$ in individual cell $j$ as a linear combination of the activities of the a priori known regulators of gene $i$ in individual cell $j$ (1). In practice, this means that we use the known targets of a transcription factor to derive its activity.

$$X_{i,j} = \sum_{k \in TFs} P_{i,k} A_{k,j} \tag{1}$$

In matrix form, *Equation 1* can be written as $X = PA$. This is an overdetermined system, meaning that there are more equations than unknowns and therefore there is no solution if all equations are linearly independent. We approximate $A$ by finding $\tilde{A}$ that minimizes $\|P\tilde{A} - X\|_2^2$. If a transcription factor has no prior targets present in $P$, we use the expression of that transcription factor as a proxy for its activity.

## Inferring regulatory interactions, single-task (Bayesian Best Subset Regression)

We utilize a bayesian best-subset regression (BBSR) method, previously described (*Greenfield et al., 2013*), for single-task network inference. At steady state, we model the expression of a gene $i$ in individual cell $j$ as a linear combination of the activities of its regulators in individual cell $j$ (2). For each gene $i$, we limit the number of potential regulators $R_i$ to the ten with the highest context likelihood of relatedness, calculated from the mutual information between all regulators and the gene $i$ (*Madar et al., 2010*), in addition to any a priori known regulator of gene $i$. Limiting the regulators is necessary before best subset regression, when we find the least squares solution to all possible combinations of predictors in set $R_i$. Because we expect a limited number of transcription factors to regulate a particular gene, our goal is to find a sparse solution for β, in which non-zero entries define both the strength and direction (activation or repression) of a regulatory relationship.

$$X_{i,j} = \sum_{k \in R_i} \beta_{i,k} A_{k,j} \tag{2}$$

Prior knowledge can be incorporated using Zellner's g-prior on the regression parameters β; in this work, we include prior interactions in the set of predictors to be modeled by best subset regression, but we do not further bias the predictors chosen with a g-prior on the regression parameters. We select the model with the lowest Bayesian Information Criterion, which adds a theoretically derived penalty term to the training error to account for model complexity and thereby reduce generalization error. After this step, the output is a matrix of inferred regression parameters β, where each entry corresponds to a regulatory relationship between transcription factor *k* and gene *i*.

## Inferring regulatory interactions, multi-task (AMuSR)

The multitask approach used here entails a joint inference of regulatory networks across multiple expression datasets. In addition to the linear assumption, in which gene expression is a linear function of the activities of regulators, we also assume that much of the underlying regulatory network is shared among related datasets (conditions). Here, we extend a previous version of the Inferelator that implements Adaptive Multiple Sparse Regression (AMuSR), which is designed to leverage cross-dataset commonalities while preserving relevant differences (*Castro et al., 2019*).

There are multiple ways of dividing the existing yeast data into multiple network data subsets, which we refer to as tasks. Within our experimental design, cells are processed and sequenced as batches, which are taken from separate environmental growth conditions. Differences between these batches are a combination of technical and biological variation. The technical variation can come from batch effect due to stress and energy-source differences associated with differing growth conditions (for example via direct effects on cell wall and thus cell lysis/yield), as well as from differences in sample preparation and sequencing. Differences in growth condition also generate biologically significant variation in gene expression due to differences in regulatory program activation. Removing technical variation while retaining biological variation through batch normalization is not feasible, and therefore these individual sample batches from separate growth conditions are taken as individual tasks for the network inference. Thus, the index 'd', below, ranges from 1 to 11 and is an index over the separate datasets corresponding to growth conditions. This separation into tasks results in the joint learning of 11 networks (one for each growth condition), followed by combination into a single global network.

Briefly, the network model is represented as a matrix **W** for each target gene (where columns are individual single-cell batches *d* and rows are potential regulators *k*) with signed entries corresponding to strength and type of regulation. We then decompose the model coefficient matrix **W** into a dataset-specific component **S** and a conserved component **B** to enable us to penalize dataset-unique and conserved interactions separately for each target gene; this separation captures differences in regulatory networks across datasets. Specifically, we apply an $l_1/l_\infty$ penalty to the **B** component to encourage similarity between network models, and an $l_1/l_1$ penalty to the other to accommodate differences to **S** (*Jalali et al., 2010*). Regularization parameters $Noentity_s$ and $Noentity_b$, representing the strength of each penalty, were chosen via Extended Bayesian Information Criterion (*Chen and Chen, 2008*). We set $Noentity_b$ to $c_b\sqrt{\frac{d \log p}{n}}$, where *d* is the number of tasks, *n* is the mean number of samples per task, and *p* is the number of predictors. We then search for $c_b$ in the log interval [0.1, 10.0] with 20 steps. We then set $Noentity_s$ such that $\frac{1}{d} < \frac{Noentity_s}{Noentity_b} < 1$, where *d* is the number of tasks and $c_s Noentity_s = Noentity_b$. We search for $c_s$ in the linear interval $[\frac{1}{d} + 0.01, 0.99]$ with 10 steps.

We can incorporate prior knowledge by using adaptive weights ($\Phi Noentity_s | l_1/l_1$) when penalizing different coefficients in the $l_1/l_1$ penalty (*Zou, 2006*). In this work, however, we chose not to bias predictors to the priors using adaptive weights, and set $\Phi l_1/l_1$ to 1. For each gene, we minimize the following function (*Castro et al., 2019*):

$$argmin_{S,B} = \sum_d || X_i^{(d)} - \hat{A}^{dT}(S_{*,d} + B_{*,d})||_2^2 + \lambda_s \sum_{k,d} |\Phi_{k,d} S_{k,d}| + \lambda_b ||B||_{1,\infty} \tag{3}$$

$$output: W = S + B$$

## Ranking interactions and data resampling

Interactions were ranked by both the overall performance of the model for each gene $i$ and the proportion of variance explained by each $\beta_{i,k}$. The output of this procedure is a matrix $S$ where $S_{i,k}$ is the confidence score on the interaction between transcription factor $k$ and gene $i$. In order to avoid overfitting and sampling biases, we repeat this procedure $N$ times by resampling the input data matrix with replacement. Finally, we rank combine the confidence scores generated by running the above inference procedure on each of the $N$ bootstrapped datasets and obtain a final matrix of combined confidence scores for the possible interactions between transcription factors (columns) and genes (rows).

## Network combination

Individual task networks were assembled into a global network by rank combining the confidence scores generated for each possible interaction between transcription factors and genes, obtaining a final matrix of combined confidence scores for the global network. Global interactions were ordered by combined confidence score, and the top interactions were kept to a threshold defined by precision = 0.5, as determined by recovery of the priors.

## Statistical analysis and differential gene expression

To analyze all growth conditions together, the raw single-cell count matrix was normalized using *multiBatchNorm* from the *scater* package in R (**McCarthy et al., 2017**). In short, this calculates size factors that are used to scale cells from different environmental condition batches so that each batch is of approximately the same mean UMI count. Cells were then library size normalized within batches and the normalized data was log-transformed with $\log_2(x+1)$ to give a transformed and normalized count matrix.

## Visualizing single cell expression data

This normalized count matrix was reduced to 50 principal components by principal component analysis (PCA) with *multiBatchPCA* from the scater package in R. *MultiBatchPCA* is standard PCA with the modification that each environmental condition batch contributes equally to the covariance matrix, even when batches are imbalanced in cell count. These principal components were projected into two dimensional space by Uniform Manifold Approximation and Projection (*UMAP*) (**McInnes et al., 2018**) and plotted.

To analyze each growth condition separately, the cells corresponding to a growth condition were selected from the raw count matrix, library-size normalized and $\log_2(x+1)$ transformed, and reduced to 50 principal components with PCA. These principal components were projected into two dimensional space by UMAP for plotting, and also used to generate a shared nearest-neighbor (sNN) graph, which is used to cluster cells using the Louvain clustering method. Each growth condition was processed and plotted separately.

## Pseudobulk differential gene expression

The raw, unmodified UMI counts of all cells from each biological replicate (with the same strain barcode) within a specific environmental growth condition were summed, resulting in 72 samples per condition (six biological replicates for each of the 12 transcription factor deletions). Summed pseudobulk expression data was then tested with DESeq2 (**Love et al., 2014**) for differential gene expression (testing against a null hypothesis of Fold Change < 1.5 and considering changes significant when $p < 0.05$ at a false discovery rate of 0.1) with no additional processing or normalization.

## Gene categorization

Cell-cycle associated genes are categorized using the Spellman annotations (**Spellman et al., 1998**). Ribosomal genes, ribosomal biogenesis genes, and induced environmental stress response genes are categorized using the Gasch annotations (**Gasch et al., 2017**). Gene category annotations are included as **Supplementary file 1**-Supplemental Table 6.

## Gene Ontology

The number of interactions was determined for each gene and each transcription factor. Interactions are considered Learned (new) if they are present in the learned network and not in the prior network; Learned (In Prior) if they are present in the learned network and not in the prior; and, Not Learned (In Prior) if they are present in the prior and not in the learned network. Each gene was mapped to Gene Ontology (GO) slim terms using the YeastGenome slim mapping (https://downloads.yeastgenome.org/curation/literature/go_slim_mapping.tab), which is a curated gene ontology mapping of high-level, broad GO terms. Interactions for all genes annotated with a GO term were summed. The generic terms 'biological_process', 'not_yet_annotated', and 'other' are removed from both target genes and regulatory transcription factors, and the common term 'transcription from RNA polymerase II promoter' was removed from regulatory transcription factors GO annotations. The 25 remaining terms with the highest number of learned (new) interactions were plotted separately for both the target genes and the regulatory transcription factors.

## Correlation plots

Gene expression data was derived experimentally in this work (FY4/FY5) or obtained from GEO. All samples are from early-log phase growth in YPD. Single-cell yeast data sets are from GSE122392 (BY4741) (*Nadal-Ribelles et al., 2019*) and GSE102475 (BY4741) (*Gasch et al., 2017*). A comparable bulk RNA control is from GSE135430 (BY4741) (*Scholes and Lewis, 2019*). Genes were ranked by expression in each cell or sample. All cells or samples from a specific experiment were rank-combined and ranks were pairwise plotted for each experiment with GGally in R.

## Variability plots

Coefficient of variation (mean over standard deviation) is calculated for each gene in each growth condition. Pearson residuals (model residual over expected standard deviation) are calculated for each gene in each cell and then the mean of the pearson residuals is taken for each growth condition. This calculation is done with the *vst* function from the sctransform package in R (*Hafemeister and Satija, 2019*). In short, this builds for each gene a regularized negative binomial model, which is then used to calculate pearson residuals for each cell compared to the model. This is done separately for each growth condition.

## Data and software availability

### Sequencing data

Raw sequencing data, the output from the cellranger pipeline to count reads, and the output from the fastqToMat0 pipeline to extract and attach genotype metadata to the count matrix are available in NCBI GEO under the accession number GEO: GSE125162.

### Single-Cell processing pipeline

The cellranger pipeline is available from 10x Genomics under the MIT license (https://github.com/10XGenomics/cellranger). The fastqToMat0 pipeline is available from GitHub (https://github.com/flatironinstitute/fastqToMat0; *Jackson, 2020*; copy archived at https://github.com/elifesciences-publications/fastqToMat0) and is released under the MIT license. Genome sequence and annotations are included as *Source code 4*.

### Network inference

The Inferelator is implemented in Python, with dependencies on the widely-distributed scientific packages Numpy (*van der Walt et al., 2011*), Scipy (*Virtanen et al., 2020*), Pandas (*McKinney, 2010*), and Scikit-learn (*Pedregosa et al., 2011*). Scaling to a high-performance computing cluster is implemented with dask (*Rocklin, 2015*). All network inference in this work was performed with the inferelator v0.3.0, using Python v3.7.3, Numpy v1.16.2, Pandas v0.24.2, Scikit-learn v0.20.3, Scipy v1.2.1, and dask v1.1.4. The inferelator package is available under the Simplified BSD licence and can be installed from PyPI (https://pypi.org/project/inferelator/) or cloned from GitHub (https://github.com/flatironinstitute/inferelator; *Jackson and Gibbs, 2020*; copy archived at https://github.com/elifesciences-publications/inferelator).

## Figure construction

*Figure 1* and *Figure 1—figure supplement 1* are constructed using Adobe Illustrator. *Figures 2–7* and accompanying supplementary figures are constructed with R. The R (v3.5.1) (*R Development Core Team, 2018*) packages used are as follows: for plotting, ggplot2 (v3.1.0) (*Wickham, 2016*), cowplot (v0.9.4) (*Wilke, 2019*), ggridges (v0.5.1) (*Wilke, 2018*), ggrastr (v0.1.7) (*Petukhov, 2019*), GGally (v1.4.0) (*Schloerke et al., 2018*), viridis (v0.5.1) (*Garnier, 2018*), RColorBrewer (v1.1–2) (*Neuwirth, 2014*), and scales (v1.0.0) (*Wickham, 2018a*); for data manipulation, dplyr (v0.7.8) (*Wickham et al., 2018*), data.table (v1.12.0) (*Dowle and Srinivasan, 2019*), reshape2 (v1.4.3) (*Wickham, 2007*), and stringr (v1.3.1) (*Wickham, 2018b*); and for single-cell analysis, scater (v1.10.1) (*McCarthy et al., 2017*), scran (v1.10.2) (*Lun et al., 2016*), umap (R: v0.2.0.0, python: v0.3.6) (*Konopka, 2018*), igraph (v1.2.2) (*Csardi and Nepusz, 2006*), DESeq2 (1.22.2) (*Love et al., 2014*), corpcor (v1.6.9) (*Schafer et al., 2017*), and sctransform (v0.2.0) (*Hafemeister and Satija, 2019*). The R scripts to generate these figures and all required data are included with *Source code 1*. Network illustrations in *Figures 6* and *7* were generated using Gephi 0.9.2 from the inferelator output network (gefx formatted); the layouts used are Force Atlas 2, Noverlap and Label Adjust. Figures were minimally modified from R outputs to enhance layout and aesthetics using Adobe Illustrator.

## Interactive figures

Interactive versions of several panels from *Figures 1–4* are available as Shiny (*Chang et al., 2018*) apps online at http://shiny.bio.nyu.edu/YeastSingleCell2019/. Source code for the Shiny app is available upon request under the MIT license.

# Acknowledgements

We would like to thank past and present members of the Gresham and Bonneau labs, as well as Christine Vogel's lab, for discussions and feedback. We thank our undergraduate researchers, especially Juli Miller, for help constructing strains. We thank Tara Rock, Olivia Micci-Smith, and Hana Husic from the NYU Genomics Core facility for troubleshooting suggestions and DNA sequencing services. RB acknowledges support from the Flatiron Institute, the Simons Foundation, the NIH (R01DK103358, R01HD096770, and R01CA229235) and the NSF (IOS1546218). DG is funded by the NIH (R01GM107466) and NSF (MCB1818234).

# Additional information

## Funding

| Funder | Grant reference number | Author |
| --- | --- | --- |
| National Institute of Diabetes and Digestive and Kidney Diseases | R01DK103358 | Richard Bonneau |
| National Institute of General Medical Sciences | R01GM107466 | David Gresham |
| National Science Foundation | MCB1818234 | David Gresham |
| Eunice Kennedy Shriver National Institute of Child Health and Human Development | R01HD096770 | Richard Bonneau |
| National Science Foundation | IOS1546218 | Richard Bonneau |
| National Cancer Institute | R01CA229235 | Richard Bonneau |
| Flatiron Institute | | Richard Bonneau |
| Simons Foundation | | Richard Bonneau |

The funders had no role in study design, data collection and interpretation, or the decision to submit the work for publication.

## Author contributions
Christopher A Jackson, Conceptualization, Resources, Data curation, Software, Formal analysis, Validation, Investigation, Visualization, Methodology, Writing - original draft, Writing - review and editing; Dayanne M Castro, Software, Formal analysis, Investigation, Visualization, Methodology, Writing - original draft, Writing - review and editing; Giuseppe-Antonio Saldi, Software, Formal analysis; Richard Bonneau, David Gresham, Conceptualization, Resources, Supervision, Funding acquisition, Project administration, Writing - review and editing

## Author ORCIDs
Christopher A Jackson [ID] https://orcid.org/0000-0002-8769-2710
Richard Bonneau [ID] https://orcid.org/0000-0003-4354-7906
David Gresham [ID] https://orcid.org/0000-0002-4028-0364

## Decision letter and Author response
Decision letter https://doi.org/10.7554/eLife.51254.sa1
Author response https://doi.org/10.7554/eLife.51254.sa2

# Additional files

## Supplementary files
• Source code 1. A 'tar.gz' archive containing R scripts used to generate *Figures 2–7* and accompanying supplementary figures with a README detailing the necessary R environment to run them locally. It also contains a data folder with the raw count matrix as a TSV file (103118_SS_Data.tsv.gz), the simulated negative data count matrix as a TSV file (110518_SS_NEG_Data.tsv.gz), a gene name metadata TSV file (yeast_gene_names.tsv), supplemental tables 5 (STable5.tsv) and 6 (STable6.tsv) as TSV files, and the yeast gene ontology slim mapping as a TAB file (go_slim_mapping.tab). *Source code 1* also contains a priors folder with the Gold Standard, the three sets of priors data tested in this work, and the YEASTRACT comparison data, all as TSV files. *Source code 1* also contains a network folder with the network learned in this paper (signed_network.tsv) as a TSV file, and the networks for each experimental condition (COND_signed_network.tsv) as 11 separate TSV files. *Source code 1* also contains an inferelator folder with the python scripts used to generate the networks for *Figures 5*, *6*, *7*.

• Source code 2. The raw count matrix as a gzipped TSV file. This file contains 38,225 observations (cells). Doublets and low-count cells have already been removed; gene expression values are unmodified transcript counts after deartifacting using UMIs (these values are directly produced by the cellranger count pipeline)

• Source code 3. The network learned in this paper as a TSV file.

• Source code 4. A '.tar.gz' archive containing the sequences used for mapping reads. It also contains a FASTA file containing the genotype-specific barcodes (bcdel_1_barcodes.fasta), a FASTA file containing the yeast S288C genome modified with markers (Saccharomyces_cerevisiae.R64-1-1.dna.toplevel.Marker.fa), and a GTF file containing the yeast gene annotations modified to include untranslated regions at the 5' and 3' end, and with markers (Saccharomyces_cerevisiae.R64-1-1.Marker.UTR.notRNA.gtf).

• Source code 5. A zipped HTML document containing the raw R output figures for *Figures 2–7* and accompanying supplementary Figures. The R markdown file to create this document is contained in *Source code 1*.

• Supplementary file 1. An excel file containing Supplemental Tables 1-6. Supplemental Table 1 contains all primer sequences used in this work. Supplemental Table 2 contains all *Saccharomyces cerevisiae* strains used in this work. Supplemental Table 3 contains all plasmids used in this work. Supplemental Table 4 contains all media formulations used in this work. Supplemental Table 5 contains the source data for modeling performance (as AUPR) that is reported graphically in *Figure 5*. Supplemental Table 6 contains the gene categorizations (cell cycle stage, RP, RiBi, etc) used in *Figure 3*.

• Transparent reporting form

## Data availability

Sequencing data has been deposited in GEO: GSE125162. Figures 2-7 (& supplementary figures) are generated from a single R markdown document. The scripts and all data necessary to do this analysis are provided as Source code 1. The raw output (knit HTML file) is provided as Source code 5. Interactive versions of several figures are available have been made available with the Shiny library in R: http://shiny.bio.nyu.edu/cj59/YeastSingleCell2019/. The Inferelator package is available on GitHub and through python package managers (i.e. pip) under an open source license (BSD).

The following dataset was generated:

| Author(s) | Year | Dataset title | Dataset URL | Database and Identifier |
|---|---|---|---|---|
| Jackson CA | 2019 | Gene regulatory network reconstruction using single-cell RNA sequencing of barcoded genotypes in diverse environments | https://www.ncbi.nlm.nih.gov/geo/query/acc.cgi?acc=GSE125162 | NCBI Gene Expression Omnibus, GSE125162 |

The following previously published datasets were used:

| Author(s) | Year | Dataset title | Dataset URL | Database and Identifier |
|---|---|---|---|---|
| Nadal-Ribelles M, Islam S, Wei W, Latorre P, Steinmetz L | 2019 | Sensitive, high-throughput single-cell RNA-Seq reveals within-clonal transcript-correlations in yeast populations | https://www.ncbi.nlm.nih.gov/geo/query/acc.cgi?acc=GSE122392 | NCBI Gene Expression Omnibus, GSE122392 |
| Gasch A | 2017 | Single-cell RNA-seq reveals intrinsic and extrinsic regulatory heterogeneity in yeast responding to stress | https://www.ncbi.nlm.nih.gov/geo/query/acc.cgi?acc=GSE102475 | NCBI Gene Expression Omnibus, GSE102475 |
| Scholes AN, Lewis JA | 2019 | Comparison of RNA Isolation Methods in Yeast on RNA-Seq: Implications for Differential Expression and Meta-Analyses | https://www.ncbi.nlm.nih.gov/geo/query/acc.cgi?acc=GSE135430 | NCBI Gene Expression Omnibus, GSE135430 |

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
