## [Decision Letter]

**Acceptance summary:**

In your article, you present individual transcriptomes of diploid yeast cells, applying your method to monitor transcriptional changes across an array of newly generated barcoded deletion mutants across a panel of stresses. Beyond the development of the method, the main and most original point of the manuscript is that you infer gene regulatory networks based on the transcriptomes retrieved from barcoded genotypes in diverse conditions. The manuscript is well written and together with related reports will become one of the golden standards for yeast single cells. You are commended for providing a complete and user-friendly dataset (deposited and interactive through a shiny app), which will be a valuable resource for the yeast community.

**Decision letter after peer review:**

Thank you for submitting your work entitled "Gene regulatory network reconstruction using single-cell RNA sequencing of barcoded genotypes in diverse environments" for consideration by *eLife*. Your article has been reviewed by three peer reviewers, and the evaluation has been overseen by a Reviewing Editor and a Senior Editor. The reviewers have opted to remain anonymous.

Our decision has been reached after consultation between the reviewers. Based on these discussions and the individual reviews below. As you will see, the reviewers found the paper to be a bit preliminary in the interpretation and the meaningfulness of the presented data. The reviewers did find the work to be potentially suitable for *eLife* and will be happy to look at a revised version, provided that you can fully address all comments raised in the individual reviews.

Reviewer #1:

Here, Jackson et al. apply for the first time a 10x Genomics protocol for scRNAseq in barcoded *S. cerevisiae* deletion strains. The work analyzed 11 transcription-factor deletion strains pooled together and responding to 11 different conditions related to nitrogen metabolism. The authors then applied a gene-regulatory network (GRN) inference method to predict a nitrogen metabolism regulatory network.

The pluses of the work are that it is a hot topic and, although not the first scRNAseq in fungi, the first to look very broadly at tens of thousands of yeast cells. The barcoding method is interesting, although unclear how it's different from the barcode sequencing that is part of the standard 10x genomics pipeline and has been used before in other systems.

The weaker points for me were in the analysis. The authors have expertise in GRN inference, but I had a hard time understanding from the main text what was novel here and specific for single cell data versus previously published methods applied to data pooled across cells/conditions. I will leave it to network inference modelers to dissect those details. But in a broader sense, I was left wanting more follow-up to show that the methods produced new insights. The authors predicted a network but as there is no biological validation and little computational validation it's unclear how big of an advance this is. I was also left wondering about the biological insights that can be gleaned from having single cell data (beyond variation in cell-cycle stage, which is readily identifiable in all scRNAseq studies). I suspect there is interesting biology in the heterogeneity in the response data but there was not much addressed on that topic. In my opinion, this is a great new method with a potentially powerful dataset. But since there are many GRN methods and this overall approach seems similar to Perturb-seq and other methods, the results and impact for me fall below the bar of *eLife*.

Some specific points are outlined below.

1) The authors report reads from 38,000 cells, which to date is the most cells studied in fungi. But the median number of genes covered is only <700. Since the paper focuses on re-bulked data (to call differentially expressed genes by DESeq2 and, I think, for their main GRN network inference?), I was left wondering how many genes are measured in the re-bulked data per condition. I was surprised how few genes were called by DESeq (Figure 4B), but it's unclear how many genes are actually measured in >1 cell condition/mutant. I was also curious what fraction of known targets (e.g. based on prior studies or ChIP-seq datasets) were called for measured genes.

2) It would be useful to know how well the 10x protocol works for cells and if the aggregated wild-type data recapitulates bulk profiles in conditions that have been previously measured. I was a little concerned at how the cells were collected, which appeared to take live cells and wash them several times in RNALater buffer – does that immediately kill cells? If not, I was wondering if that is inducing a response. I was also left wondering if the protocol captures only the most abundant transcripts. Perhaps I missed this on the supplement, but I was wondering how the% cells in which a transcript was measured compared to RPKM from bulk measurements. Clearly more abundant transcripts will be more easily captured, but some more analysis here would be useful for a new method.

3) The GRN modeling was not clear to me from the main text. It appears that the authors are using their published Inferelator method that takes priors based on ChIP-seq data, and re-bulks the scRNAseq data (at least for the multi-task inference). Perhaps their point is that the 10x approach allows pooling of many genotypes and conditions, but for me the analysis missed the potential power of having single cell data. The authors make statements on the networks in Figure 6 and Figure 7 about the number of "novel" regulatory connections – but I saw no validation of those predictions, including by computation. How do we know that any of these are real and that the method is producing new insights? AUROCs comparing to known data is not enough to say that new regulatory connections were discovered. This was especially true for Figure 7 – how do we know these new predictions mean anything about a connection to cell cycle without some validation?

4) I had some quibbles with part of the Discussion.

i) First, while the authors cite several recent *S. cerevisiae* scRNAseq datasets, saying that this is the "first report of large-scale scRNAseq" in yeast is not accurate – it's true they measured 38,000 cells but at a depth of only <700 genes per cell, which is far fewer than 2500-3000 of other studies in several hundred cells. A fairer sentence is required, also citations of recent *Sz. pombe* scRNAseq (Saint et al., 2019) should be included.

ii) "We observe significant heterogeneity in individual cells.… Much of this variation can be explained by the mitotic cell cycle" – that statement is not true, there does not seem to be an attempt to quantify heterogeneity over most genes. That they see heterogeneity in cell cycle stage as expected does not mean that cell cycle stage explains heterogeneity in the rest of the response, which was not reported on here.

Reviewer #2:

In the manuscript submitted by Jackson et al., the authors proposed to profile individual transcriptomes of diploid yeast cells optimizing the 10x genomics pipeline. The manuscript demonstrates the feasibility and robustness of the method and the authors applied it to monitor the transcriptional networks across an array of newly generated barcoded deletion mutants across a panel of stresses. The manuscript is scientifically sound and of outstanding quality, it's well written and together with previous reports this year will become one of the golden standards for yeast single cells.

The authors provide a complete and user-friendly dataset (deposited and interactive through a shiny app) that will be a valuable resource for the yeast community. All in all, I think this is a very solid manuscript that with minor corrections I would strongly support for publication in *eLife*.

Beyond the development of the method, the main and most original point of the manuscript is that the authors aim to generate infer gene regulatory networks based on the transcriptomes retrieved from barcoded genotypes in diverse conditions. I have some questions and comments (of varying levels of concern) that I feel should be addressed in the current version of the manuscript.

The authors leverage in their previous experience in GRN reconstruction and propose this approach has allowed them to discover novel regulatory relationships between cell cycle-regulated gene expression in response to changes in nitrogen source. In my opinion, this part of the manuscript needs to be reinforced and some of the conclusions driven from the GRN should be and some representative novel regulatory relationships experimentally demonstrated. As well the authors should provide context to their findings as they often read a bit disconnected.

In terms of data quality, the number of mitochondrial and ribosomal reads per genotype and condition should be plotted, as a quality metric and given that a lot of ribosomal gene expression is cell cycle regulated. This could be relevant to understand separate clusters within conditions which the authors do not mention in the results and/or discussion and given the fact that different zymolyase concentrations were used for cell lysis.

The optical density of the cells at the harvesting (besides the total cell number) should be provided to ease the reproducibility for other labs.

If I understand correctly, cell cycle clusters within conditions in Figure 3 is confusing. For example expression of DSE2 or PIR1 seem to be highest in the green and grey clusters respectively (Figure 3A) however in Figure 3B panel I the highest expression is assigned to the grey-yellow for DSE2 and PIR1 to the green cluster.

As well, the UMAPS from Figure 3A, the authors claim the clustering is mainly condition-dependent and genotype-independent. However, conditions like MMEtOH, CSTARVE, NLIM-PRO have clear clusters that do not seem cell cycle-dependent and these might be biologically relevant. The authors should at least comment or those or run a DE analysis to see what these are.

How do the newly generated data compare to Gasch et al., 2017 and Nadal-Ribelles et al., 2019?

The number of differentially expressed genes even in the YPD condition seems a bit low. How do these pseudobulk compare to the deletome data (Holstege lab) or other published datasets?

As well, the authors use DESeq in Figure 4B, but these results are contradictory with what is shown in Figure S4Bi which is done by Welch testing. This is a bit confusing and does not add much to the reader. I would suggest to run DE between conditions using DESeq or provide the reasoning as to why these two different approaches are used and done?

Why do transcription factor activities FKH1 and FKH2 and SWI4 SWI5 do not overlap (they almost seem mutually exclusive), one would expect them to have similar profiles. Similarly, NDD1 regulates S-phase genes but the TFA does not overlap with the HTB expression shown in Figure 3.

Reviewer #3:

This paper by Gresham, Bonneau and colleagues presents to date the largest single cell RNA-seq datasets in yeast using a novel deletion and barcoding strategy that enables them to measure individual cells under different genetic perturbations of transcription factors and environmental conditions. The study is focused on better understanding the regulatory network in nitrogen starvation however it could be broadly applied across multiple conditions. Some findings are: the genotypes tend to be generally uniformly present in all conditions except RTG1 and 3 and GLN3; there is a co-regulated set of genes regulated by cell cycle and nitrogen TFs; multi-task learning is a viable approach for network inference in scRNAseq data. The paper is a significant contribution to the field providing a novel dataset to yeast and general gene regulation community.

I have some comments, which I think are minor and can strengthen the messages of the paper.

1) In Figure 5C, is the single task network inferred by merging all the data and learning a single network or by learning separate networks with single tasking and aggregating the results? If not, how does the single task per condition followed by aggregation perform?

2) The authors don't get much into the context-specificity of the inferred networks. They interpret only the final aggregated network. It would be useful to know how similar the individual condition-specific networks are and if there is a conserved core used by multiple conditions. The only comparison of context specificity is being done at the level of AUPR, it might be helpful to do this comparison just by comparing the inferred networks.

3) Some discussion about the variation in the AUPRs would be helpful. Is it because the gold standard is biased towards the conditions on which the AUPR is high. It seems the AUPR for the MMEtOH network is close to what is inferred by the multi-task learning, and some explanation of why this might be is helpful.

4) A comparison of the AUPR of the single task condition-specific networks and the multi-task condition specific network could further show the advantage of the multi-task learning framework.

5) It would be helpful to emphasize if and how the multi-task learning approach used here was from extended from the Castro, 2019 paper.

6) The Discussion could be strengthened. The authors present some results about the interplay between cell cycle and nitrogen response. It was not clear why this is interesting to study beyond that there is a shared regulatory program. This might be worth bringing up in the Discussion to tie back to the initial goal of inferring a network for nitrogen metabolism and the TOR signaling pathway and the general role of cell cycle and stress response.

7) The authors don't find a substantial impact of TF knockout on gene expression under different conditions (I assume the comparisons were done while controlling for the conditions). How does this compare to bulk data? How much of this observation could be due to the sparsity of scRNAseq data versus the redundancy of TFs.

---

## [Author Response]

We have revised the manuscript based on the feedback that we have received from reviewers. To address the reviewer concerns we have undertaken additional experimentation and analyses. Below, we summarize the major changes made to the text.

1) A common concern of the three reviewers was the quality and utility of the single-cell RNAseq data. To address this concern we performed bulk RNAseq on wild-type cells from the YPD rich media condition. Our analysis of this experiment used the same computational pipeline as used for our single-cell reads and indicates that scRNAseq in yeast produces data that are very comparable to that produced from bulk RNAseq. In addition, we have performed comparisons to other published single-cell data and to another bulk sequencing experiment from GEO, which support this conclusion. This analysis is included as Figure 2—figure supplement 2. We have added additional QC metrics suggested by the reviewers as Figure 2—figure supplement 3. We believe that these results validate the quality of data produced using this method and obviate any concerns that our sample processing introduces biases in the data.

2) A second common concern was validation of the network predictions made by our inference technique. To address this concern, we have compared the novel predictions made in this work to the YEASTRACT database. More than half of the predictions we make are supported by some type of evidence (either changes in target gene expression or physical localization by ChIP). As this evidence was not included in the modeling priors we believe that this provides orthogonal support for at least 50% of the interactions that we discovered in our study.

3) To address concerns regarding the computational approaches, we have regenerated the gene regulatory network results reported in Figures 5-7 incorporating the following changes:

a) We updated the Inferelator package for python to version 0.3.0 (some minor software bugs were fixed and the package is now compatible with Python 3; this upgrade did not change any modeling results).

b) We updated the YEASTRACT prior data to match the 2019 release of the YEASTRACT database; as a result, ~2000 TF-gene interactions were removed as being poorly supported, and ~3500 TF-gene interactions were added relative to the 2018 release used in our initial submission. Updating the modeling priors with these changes improved many of our modeling results.

c) While preparing this revision, we identified an issue that was causing our network inference to make spurious predictions due to genes that have a mean count value in at least one task group that is very close to zero (but not zero). We have mitigated this problem by filtering out genes that don’t have a minimum mean count of 0.05 (at least 1 read per 20 cells). For multi-task learning, this filter is applied task-wise; genes filtered from one task are still modeled in other tasks. This approach results in removal of the spurious interactions.

d)We have performed additional cross-validation for Figure 5B (now 20 runs).

4) The Materials and methods section has been substantially updated to incorporate these additions to the manuscript and several minor typos were corrected.

5) One reviewer identified a figure which had a labeling error introduced during panelling with Adobe Illustrator. This motivated us to include as a supplemental data file (Source code 5) an HTML document generated with Rmarkdown that contains Figures 2-7 (including all figure supplements). The figures produced in this way have not been subject to any subsequent adjustments in Illustrator and thus provide a reproducible record of the data analysis and presentation that can be regenerated by running the Rmarkdown script (which is included in Source code 1).

Reviewer #1:[…] The pluses of the work are that it is a hot topic and, although not the first scRNAseq in fungi, the first to look very broadly at tens of thousands of yeast cells. The barcoding method is interesting, although unclear how it's different from the barcode sequencing that is part of the standard 10x genomics pipeline and has been used before in other systems.The weaker points for me were in the analysis. The authors have expertise in GRN inference, but I had a hard time understanding from the main text what was novel here and specific for single cell data versus previously published methods applied to data pooled across cells/conditions. I will leave it to network inference modelers to dissect those details. But in a broader sense, I was left wanting more follow-up to show that the methods produced new insights. The authors predicted a network but as there is no biological validation and little computational validation it's unclear how big of an advance this is. I was also left wondering about the biological insights that can be gleaned from having single cell data (beyond variation in cell-cycle stage, which is readily identifiable in all scRNAseq studies). I suspect there is interesting biology in the heterogeneity in the response data but there was not much addressed on that topic. In my opinion, this is a great new method with a potentially powerful dataset. But since there are many GRN methods and this overall approach seems similar to Perturb-seq and other methods, the results and impact for me fall below the bar of eLife.

We thank the reviewer for recognizing the novelty of our study. As we emphasize in our manuscript, adopting the 10x Genomics systems for budding yeast (and microbes in general) is non-trivial and we have developed a robust methodology that we are convinced will be widely adopted. We are aware that the Perturb-seq method does express a barcode that is informative about the gRNA for CRISPR/Cas9 screens; however, our method expresses a barcode from the drug resistant cassette that is used for deleting the gene of interest and thus provides a more direct readout of the cells’ genotypes. The successful application of 10x Genomics to microbial cells and the method of multiplexing genotypes make this paper of broad interest to the readers of *eLife*.

With respect to the novelty of the computational approaches, we concur that there are indeed many network inference methods. However, we are convinced that there is a great deal of novelty in applying GRN inference techniques to real-world single-cell data in a testable way. We are aware of only one study that is similar in scope and goals to ours; it was released as a preprint around the same time as this work, and it has been recently accepted for publication in Science. (Norman, T.M., et al. (2019). Exploring genetic interaction manifolds constructed from rich single-cell phenotypes). The rigorous testing of unsolved aspects of GRN using scRNAseq data, such as imputation, and the application of multitask learning are key features of our manuscript that we are convinced will be of great value to the larger biological research community that aims to use scRNAseq for GRN reconstruction.

Some specific points are outlined below.1) The authors report reads from 38,000 cells, which to date is the most cells studied in fungi. But the median number of genes covered is only <700. Since the paper focuses on re-bulked data (to call differentially expressed genes by DESeq2 and, I think, for their main GRN network inference?), I was left wondering how many genes are measured in the re-bulked data per condition. I was surprised how few genes were called by DESeq (Figure 4B), but it's unclear how many genes are actually measured in >1 cell condition/mutant. I was also curious what fraction of known targets (e.g. based on prior studies or ChIP-seq datasets) were called for measured genes.

It is critical to note that only Figure 4B, Figure 4C, and Figure 4—figure supplement 1B rely on bulked data. All other figures and analyses in our paper are based on single-cell data. The huge increase in the number of individual measurements that scRNAseq provides is particularly important for the GRN inference. We have clarified this point in the manuscript.

Cells cultured in YPD, the most commonly used rich media, are the best baseline for comparison to bulk methods. We have 11,037 cells present in this data set. A median of 684 genes are detected per cell, and 5,533 genes have at least one read across all cells (of 5,773 protein-coding genes). 5,403 genes are counted in more than 1 cell in this condition. As additional points of comparisons, 299 genes average at least one count/cell, and 1,831 genes average at least one count in every 10 cells. We believe that the low count/cell is the result of overall low sampling rates for transcripts (some of which may be technically addressable in the future), and is not the result of specific gene bias, which is supported by the high correlation to our new data using trizol extracted bulk RNAseq, which is included as Figure 2—figure supplement 2.

With respect to the fraction of known targets that are measured we compared the novel interactions identified in our study and find that about 50% of them are supported by prior studies included in the YEASTRACT database.

2) It would be useful to know how well the 10x protocol works for cells and if the aggregated wild-type data recapitulates bulk profiles in conditions that have been previously measured. I was a little concerned at how the cells were collected, which appeared to take live cells and wash them several times in RNALater buffer – does that immediately kill cells? If not, I was wondering if that is inducing a response. I was also left wondering if the protocol captures only the most abundant transcripts. Perhaps I missed this on the supplement, but I was wondering how the% cells in which a transcript was measured compared to RPKM from bulk measurements. Clearly more abundant transcripts will be more easily captured, but some more analysis here would be useful for a new method.

We thank the reviewer for raising this point; this is a very important methodological detail. It is necessary to fix cells prior to processing to prevent changes in the transcriptome. We chose to use RNAlater (saturated ammonium sulfate) in the initial development of our protocol as it is a widely used approach that has been successfully applied to yeast in the past. Snap freezing in liquid nitrogen is not a fixative but a storage solution, and it is not feasible with the single-cell workflow anyway, and we found in preliminary testing that methanol fixation introduced obvious bias after reverse transcription.

To specifically address the reviewer’s concern about transcript capture bias, we performed an RNA sequencing experiment using bulk RNA extracted from wildtype cells in YPD using a trizol-based protocol. For this bulk experiment, we used a 3’ end barcoding with UMI strategy that is directly comparable to the 10x genomics method for cDNA synthesis and library preparation, and which can be analyzed through the same computational pipeline as our single-cell data. The results of this experiment are presented in Figure 2—figure supplement 2. We find that, in aggregate, the single-cell expression data from WT cells in YPD correlates very well with the expression data from bulk RNA (spearman correlation of 0.94). In addition, despite considerable differences in technical protocols, as well as using different strains (auxotrophic haploids vs prototrophic diploids in our study), we find that the single-cell expression data from our 10x Genomics protocol also correlates well with the expression data from the yscRNAseq protocol reported by Nadal-Ribelles et al, 2019 (spearman correlation of 0.83). We believe that this convincingly shows that the 10x Genomics-based method we developed for scRNAseq does not introduce any biases in the data.

3) The GRN modeling was not clear to me from the main text. It appears that the authors are using their published Inferelator method that takes priors based on ChIP-seq data, and re-bulks the scRNAseq data (at least for the multi-task inference). Perhaps their point is that the 10x approach allows pooling of many genotypes and conditions, but for me the analysis missed the potential power of having single cell data. The authors make statements on the networks in Figure 6 and Figure 7 about the number of "novel" regulatory connections – but I saw no validation of those predictions, including by computation. How do we know that any of these are real and that the method is producing new insights? AUROCs comparing to known data is not enough to say that new regulatory connections were discovered. This was especially true for Figure 7 – how do we know these new predictions mean anything about a connection to cell cycle without some validation?

We have clarified in the text that all network modeling is based on single-cell data and does not use pseudo-bulked data. We now emphasize that a novel interaction in this context is defined as interactions we discover using our network inference method without prior knowledge. To validate these interactions we now include a comparison to data in the YEASTRACT database that was not included in our network inference method as priors. Of the 6114 novel interactions discovered in our network, YEASTRACT provides evidence of an existing regulatory relationship for more than half. We believe that this provides orthogonal validation of our newly discovered interactions.

As our data allows us to determine the expression of individual cells the true advantage is accessing expression at the single-cell level. While some high-quality work has been done to study the intersection of cell-cycle regulation and metabolic regulation, we believe that studying asynchronous cultures at the single-cell level will be immensely valuable. Studying these novel regulatory relationships will be the subject of future work.

4) I had some quibbles with part of the Discussion.i) First, while the authors cite several recent *S. cerevisiae* scRNAseq datasets, saying that this is the "first report of large-scale scRNAseq" in yeast is not accurate – it's true they measured 38,000 cells but at a depth of only <700 genes per cell, which is far fewer than 2500-3000 of other studies in several hundred cells. A fairer sentence is required, also citations of recent Sz. pombe scRNAseq (Saint et al., 2019) should be included.

In response to this comment we have changed the phrase ‘large-scale’ to ‘droplet-based’ and changed instances of ‘yeast’ in this context to ‘budding yeast’. We have also noted that other work has higher read depth per cell than this work and cite the recent study by Saint et al., 2019.

ii) "We observe significant heterogeneity in individual cells.… Much of this variation can be explained by the mitotic cell cycle" – that statement is not true, there does not seem to be an attempt to quantify heterogeneity over most genes. That they see heterogeneity in cell cycle stage as expected does not mean that cell cycle stage explains heterogeneity in the rest of the response, which was not reported on here.

We provide plots of descriptive measures of variability as Figure 2—figure supplement 4. Our experimental design does not allow for a biological interpretation of these results as we are currently not able to distinguish technical variability from biological variability on a per-gene basis. We have included additional discussion about the potential for heterogeneity in some growth conditions; Figure 3—figure supplement 2 suggests that there are some cells undergoing different responses to stressful conditions. Although we are cautious about the capacity for our experimental to distinguish between biological heterogeneity and technical heterogeneity on a per-gene basis we provide these results as an additional example of the potential power of our approach.

Reviewer #2:[…] The authors provide a complete and user-friendly dataset (deposited and interactive through a shiny app) that will be a valuable resource for the yeast community. All in all, I think this is a very solid manuscript that with minor corrections I would strongly support for publication in eLife.Beyond the development of the method, the main and most original point of the manuscript is that the authors aim to generate infer gene regulatory networks based on the transcriptomes retrieved from barcoded genotypes in diverse conditions. I have some questions and comments (of varying levels of concern) that I feel should be addressed in the current version of the manuscript.The authors leverage in their previous experience in GRN reconstruction and propose this approach has allowed them to discover novel regulatory relationships between cell cycle-regulated gene expression in response to changes in nitrogen source. In my opinion, this part of the manuscript needs to be reinforced and some of the conclusions driven from the GRN should be and some representative novel regulatory relationships experimentally demonstrated. As well the authors should provide context to their findings as they often read a bit disconnected.

We now include a comparison of the network which we have learned to regulatory interactions defined in the YEASTRACT database that we did not provide to our network inference method as priors. Of the 6,114 new interactions identified in our network, more than half have evidence of an existing regulatory relationship in YEASTRACT. While additional experimentation would certainly provide additional evidence for individual interactions, we feel that this result validates the overall network. Dissection of individual interactions and further functional characterization of novel regulatory relationships between the cell cycle and nitrogen metabolism is the subject of future work.

In terms of data quality, the number of mitochondrial and ribosomal reads per genotype and condition should be plotted, as a quality metric and given that a lot of ribosomal gene expression is cell cycle regulated. This could be relevant to understand separate clusters within conditions which the authors do not mention in the results and/or discussion and given the fact that different zymolyase concentrations were used for cell lysis.

This is an excellent suggestion. We performed this analysis and have added Figure 2—figure supplement 3 which contains total counts, ribosomal genes, ribosomal biogenesis genes, induced environmental stress response genes, and mitochondrial genome genes overlaid over the UMAP plots.

The optical density of the cells at the harvesting (besides the total cell number) should be provided to ease the reproducibility for other labs.

Cell concentrations were determined by cell counting (not OD); we have revised the Materials and methods section to include the concentrations at harvest (the DIAUXY cell density is not available; cells were harvested based on glucose concentration in media and the cell density was only measured after resuspension in RNAlater. Back calculation from this value suggests a culture density of ~1.0 x 10^8^ cells/mL).

If I understand correctly, cell cycle clusters within conditions in Figure 3 is confusing. For example expression of DSE2 or PIR1 seem to be highest in the green and grey clusters respectively (Figure 3A) however in Figure 3B panel I the highest expression is assigned to the grey-yellow for DSE2 and PIR1 to the green cluster.

We thank the reviewer for pointing out this mistake. The labeling on Figure 3B was incorrect; the DSE2 and PIR1 labels were swapped (they were correct in Figure 3—figure supplement 1B). This error has been fixed.

As well, the UMAPS from Figure 3A, the authors claim the clustering is mainly condition-dependent and genotype-independent. However, conditions like MMEtOH, CSTARVE, NLIM-PRO have clear clusters that do not seem cell cycle-dependent and these might be biologically relevant. The authors should at least comment or those or run a DE analysis to see what these are.

This is an excellent suggestion. We have relabeled Figure 3A with expression of gene categories and included it as Figure 3—figure supplement 2. It appears that several stressful conditions have clusters that are upregulated for environmental stress response genes and downregulated for ribosomal and cell-cycle genes. The most likely explanation is that in higher-stress conditions, we are observing some cells that are in a quiescent state. We propose this possibility in the main text while cautioning that the static nature of our experimental design limits our ability to conclusively demonstrate this.

How do the newly generated data compare to Gasch et al., 2017 and Nadal-Ribelles et al., 2019?

We have compared our scRNAseq data to several published expression datasets in Figure 2—figure supplement 2. We find reasonable agreement with Nadal-Ribelles et al., 2019 despite differences in strain and experimental technique. The agreement with data form Gasch et al., 2017 is less similar, likely reflecting differences in genetic background, growth conditions, and experimental technique. These comparisons highlight the challenge of integrating single-cell data from separate experiments. However, several recent papers have explored integration of single-cell data (e.g. Butler et al., 2018, Stuart et al., 2019). One of the main long-term advantages of our work (that we will explore in the future) is that it could be applied to integrate multiple data sets that have been prepared differently into unified network models.

The number of differentially expressed genes even in the YPD condition seems a bit low. How do these pseudobulk compare to the deletome data (Holstege lab) or other published datasets?

The deletome data from Holstege 2014 was generated in synthetic complete media (the closest comparison in our data set is minimal media). The TFs which we have focused on are in pathways known to be dysregulated in auxotrophic strains like the BY4741 family used as a basis for the yeast deletion collection (the auxotrophies are all deletions in nitrogen anabolic pathways; our work does not use these strains, which makes direct comparisons to the standard yeast deletion collection more challenging).

As well, the authors use DESeq in Figure 4B, but these results are contradictory with what is shown in Figure S4Bi which is done by Welch testing. This is a bit confusing and does not add much to the reader. I would suggest to run DE between conditions using DESeq or provide the reasoning as to why these two different approaches are used and done?

We agree with the reviewer that this analysis adds nothing to this work and is a source of confusion. Therefore, we have removed the t-test based figures.

Why do transcription factor activities FKH1 and FKH2 and SWI4 SWI5 do not overlap (they almost seem mutually exclusive), one would expect them to have similar profiles. Similarly, NDD1 regulates S-phase genes but the TFA does not overlap with the HTB expression shown in Figure 3.

We thank the reviewer for identifying this inconsistency. While looking into this, we identified an issue with the modeling that disproportionately affected genes with extremely low average values in one modeling task resulting in them highly influencing certain TF activities. We have corrected this problem by adding a filter to remove genes with low average expression, and we have replaced NDD1 in this figure with MBP1, another cell-cycle TF.

SWI4 and SWI5 are not currently annotated on SGD as acting in the same stage of the cell cycle (SWI4 is annotated as late G1/S and SWI5 is annotated as early G1/M), and so we don’t expect them to overlap. TFs with functional redundancy like FHK1 and FHK2 would be expected to overlap, but the redundancy makes modeling regulatory relationships more challenging (this problem is one of the main reasons we have chosen nitrogen metabolism as our model, as it is also full of functional redundancies and cross-regulatory loops). Methodological approaches to functional redundancies is the subject of ongoing work.

Reviewer #3:[…] I have some comments, which I think can strengthen the messages of the paper.1) In Figure 5C, is the single task network inferred by merging all the data and learning a single network or by learning separate networks with single tasking and aggregating the results? If not, how does the single task per condition followed by aggregation perform?4) A comparison of the AUPR of the single task condition-specific networks and the multi-task condition specific network could further show the advantage of the multi-task learning framework.

Both of these (point 1 and point 4) are excellent suggestions. Initially, Figure 5D was comparing merged data learning a single network to AMuSR. Figure 5D now compares all data combined to learn a single-task network [BBSR (ALL)], all data learned separately with single-tasking and then aggregated into a single network [BBSR (BY TASK)], and all data which is learned together with multitask learning [AMuSR (MTL)]. In addition, the second panel of 5D has been replaced with a comparison of the task-specific networks learned individually with BBSR (labeled as [BBSR (BY TASK)]), and the task-specific networks learned jointly with AMuSR (labeled as [AMuSR (MTL)]). We find that the majority of the performance gained by MTL is due to separating individual conditions; sharing information during regression improves performance on some task-specific networks, but decreases performance on others (overall the effect balances out on the final aggregate network).

2) The authors don't get much into the context-specificity of the inferred networks. They interpret only the final aggregated network. It would be useful to know how similar the individual condition-specific networks are and if there is a conserved core used by multiple conditions. The only comparison of context specificity is being done at the level of AUPR, it might be helpful to do this comparison just by comparing the inferred networks.

This is an excellent point. The composition of the final, aggregate network is examined in Figure 6—figure supplement 1C-E. We find that there does exist a common core of interaction which we recover in every task, most of which are in the prior. Much of the learned network corresponds to interactions found in multiple condition-specific networks, but we also find a large number of interactions that are unique to a single condition-specific network (when we examine individual condition networks we find a large number of TF-gene interactions which are present in only one network; most of these are not included in the final, aggregate network, but some are). We think that this observation provides a strong justification for learning networks from separate growth conditions with distinct transcriptional profiles and have emphasized this point in our revised manuscript.

3) Some discussion about the variation in the AUPRs would be helpful. Is it because the gold standard is biased towards the conditions on which the AUPR is high. It seems the AUPR for the MMEtOH network is close to what is inferred by the multi-task learning, and some explanation of why this might be is helpful.

This is a great point. We think that the variation is mainly due to cells in certain conditions requiring more of their transcriptome to be active. Cells in rich YPD media have mostly glycolytic, cell growth, and cell cycle expression programs active, whereas cells in minimal MMD media require all of these processes plus anabolic pathways for a number of essential compounds. Cells in MMEtOH can turn off glycolysis but need respiration and redox-related pathways to be active. We have added this interpretation to the main text.

5) It would be helpful to emphasize if and how the multi-task learning approach used here was from extended from the Castro, 2019 paper.

Although there are some changes in implementation, the multi-task learning approach here is the same conceptually as the technique introduced in Castro et al., 2019. The most important difference is that we’ve chosen not to weight prior interactions differently than interactions that are not in the prior during modeling; priors in this work are only used to calculate transcription factor activity. We have clarified this in the text.

6) The Discussion could be strengthened. The authors present some results about the interplay between cell cycle and nitrogen response. It was not clear why this is interesting to study beyond that there is a shared regulatory program. This might be worth bringing up in the Discussion to tie back to the initial goal of inferring a network for nitrogen metabolism and the TOR signaling pathway and the general role of cell cycle and stress response.

We have added a paragraph to our Discussion to more explicitly discuss the relationship between nitrogen metabolism, TOR signaling and the cell cycle.

7) The authors don't find a substantial impact of TF knockout on gene expression under different conditions (I assume the comparisons were done while controlling for the conditions). How does this compare to bulk data? How much of this observation could be due to the sparsity of scRNAseq data versus the redundancy of TFs.

Our new experiments show scRNAseq data is in good agreement with bulk RNAseq data. We think that the limited impact of TF knockouts on expression is consistent with earlier published studied (e.g. the Holstege deletome study). Therefore, we think that the primary limitations of TF deletions are 1) functional redundancy and 2) studying expression in static rather than dynamic conditions. We now include these points in our Discussion and propose alternative approaches that might elicit stronger transcriptional responses (e.g. inducible overexpression of TFs).